# Velocity coupling of grid cell modules enables stable embedding of a low dimensional variable in a high dimensional neural attractor

**Noga Mosheiff[1], Yoram Burak[1,2]***

[1]Racah Institute of Physics, Hebrew University, Jerusalem, Israel; [2]Edmond and Lily Safra Center for Brain Sciences, Hebrew University, Jerusalem, Israel

**Abstract** Grid cells in the medial entorhinal cortex (MEC) encode position using a distributed representation across multiple neural populations (modules), each possessing a distinct spatial scale. The modular structure of the representation confers the grid cell neural code with large capacity. Yet, the modularity poses significant challenges for the neural circuitry that maintains the representation, and updates it based on self motion. Small incompatible drifts in different modules, driven by noise, can rapidly lead to large, abrupt shifts in the represented position, resulting in catastrophic readout errors. Here, we propose a theoretical model of coupled modules. The coupling suppresses incompatible drifts, allowing for a stable embedding of a two-dimensional variable (position) in a higher dimensional neural attractor, while preserving the large capacity. We propose that coupling of this type may be implemented by recurrent synaptic connectivity within the MEC with a relatively simple and biologically plausible structure.

DOI: https://doi.org/10.7554/eLife.48494.001

## Introduction

Much of the research on neural coding in the brain is focused on sensory representations, which are driven by external inputs that can be precisely controlled experimentally. In comparison, less is known about neural coding in deep brain structures, in which the dynamics reflect the outcome of internal computations. A notable exception is the hippocampal formation, where neural activity has been linked to high level cognitive variables such as an animal's estimate of its position within its environment (*O'Keefe and Nadel, 1978*; *Moser et al., 2008*; *Taube et al., 1990*), or its estimate of elapsed time within a trial of a trained behavioral task (*Manns et al., 2007*; *Pastalkova et al., 2008*; *Itskov et al., 2011*; *Eichenbaum, 2014*).

Specifically, the representation of position by grid cells (*Hafting et al., 2005*) in the medial entorhinal cortex (MEC) has led to new, unexpected insights on the neural coding of such quantities: even though position is a low dimensional variable, it is jointly encoded by several distinct populations of cells (modules: *Stensola et al., 2012*), exhibiting periodic spatial responses with varying spatial scales. The spatial responses of all grid cells within a module are characterized by the same grid spacing and orientation, while differing from each other only by a rigid translation. The representation of position by each module is ambiguous, but taken together, the joint activity in several modules constitutes a highly efficient and unambiguous neural code (see *Burak, 2014*). Due to its distributed organization, the grid cell code possesses a high dynamic range (ratio between the range of unambiguous representation and resolution; *Burak, 2014*), greatly exceeding the performance of unimodal coding schemes such as the representation of position by place cells in the hippocampus

**\*For correspondence:**
yoram.burak@elsc.huji.ac.il

**Competing interests:** The authors declare that no competing interests exist.

(*Fiete et al., 2008*; *Sreenivasan and Fiete, 2011*; *Mathis et al., 2012b*; *Wei et al., 2015*; *Mosheiff et al., 2017*).

Alongside the potential benefits arising from the combinatorial nature of the grid cell code, the distributed representation of position over several modules and spatial scales poses a mechanistic challenge to the underlying neural circuitry. The difficulty lies in the hypothesized role of the hippocampal formation, and specifically the MEC, in maintenance of short-term memory and idiothetic path integration, as opposed to pure representation. When grid cells update their activity, for example based on self motion, they must do so in a coordinated manner, in order for them to coherently represent a position in two-dimensional space, a variable of much lower dimensionality than the joint activity of all cells.

Neurons within a module maintain strict relationships in their joint activity (*Yoon et al., 2013*). These relationships are maintained across environments (*Fyhn et al., 2007*); under abrupt distortions of the environment (*Barry et al., 2007*; *Yoon et al., 2013*); in novel environments (*Barry et al., 2012*; *Yoon et al., 2013*), in which stable place fields are absent; during hippocampal inactivation (*Almog et al., 2019*); under genetic perturbations that disrupt the spatial periodicity in the response of individual cells (*Allen et al., 2014*); and in sleep (*Trettel et al., 2019*; *Gardner et al., 2019*). The rigidity of the correlation structure strongly suggests that the neural activity within a module is tightly coordinated by recurrent connectivity, consistent with attractor models of grid cell activity (*McNaughton et al., 2006*; *Fuhs and Touretzky, 2006*; *Guanella et al., 2007*; *Burak and Fiete, 2009*), which propose that synaptic connectivity restricts the joint activity within a module to lie on a two-dimensional manifold. Additional support for attractor models has been recently obtained by imaging activity of multiple grid cells using calcium imaging in rats running on a virtual one-dimensional track (*Heys et al., 2014*; *Gu et al., 2018*). These studies revealed a relationship between position on the cortical sheet and the preferred firing locations of grid cells, as predicted by the variants of attractor models that rely on distance-dependent connectivity.

In contrast to the strong correlation in the activity of neurons within a module, much less is known about coupling of neurons that belong to different modules. A network of grid cells organized in $m$ modules, each independently structured as a two-dimensional continuous attractor, possesses a $2m$ dimensional space of accessible steady states. Yet at any given time, continuous motion of the animal corresponds to a two-dimensional subspace of the possible local changes in the state of the $m$ modules. Considering that noise may corrupt the representation of position in each module separately, the maintenance of a coherent representation of position across modules necessitates some form of coupling between them (*Welinder et al., 2008*; *Sreenivasan and Fiete, 2011*; *Burak, 2014*). *Figure 1A* demonstrates the need for this coupling: incoherent drifts in the positions represented by different modules, accrued due to noise, can rapidly produce a joint representation of position that is incompatible with any position in the close vicinity of the animal (*Fiete et al., 2008*; *Welinder et al., 2008*; *Burak, 2014*; *Vágó and Ujfalussy, 2018*). The desired coupling across modules is more subtle than the one observed within a module: the coupling should restrict *changes* in the states of different modules to lie within the two-dimensional sub-space that corresponds to smooth movement of the animal within its local environment. However, to preserve the high dynamic range of the code, the coupling should not restrict the dimensionality of the accessible steady states.

To further illustrate this point, it is instructive to consider an analogy of grid cell coding to the representation of a one-dimensional position by the rotation angles $\theta_1$ and $\theta_2$ of two meshing gears (*Figure 1B*). We imagine that motion along the one-dimensional axis corresponds to coordinated rotation of the two gears (*Figure 1B*, bottom). If the radii $R_1$ and $R_2$ of the two gears are incommensurate, a large distance is traversed before the two meshing gears come close to a previously visited state, thus allowing for a large range of positions to be unambiguously represented. However, it is crucial in this scheme that during continuous motion, the gears rotate in a coordinated manner: $\dot{\theta}_1 R_1 = \dot{\theta}_2 R_2$. This relationship between the phase velocities $\dot{\theta}_1$ and $\dot{\theta}_2$ is enforced by the meshing cogs along the circumference of the two gears. In the absence of this mechanical constraint, small movement of one gear relative to the other can abruptly transport the represented position to a distant location, unrelated to the original position. Note that the absolute angles of the two meshing gears are not constrained: in fact, the large capacity of the representation relies on the fact that any combination of the two angles is accessible (compare panels B-C in *Figure 1*).

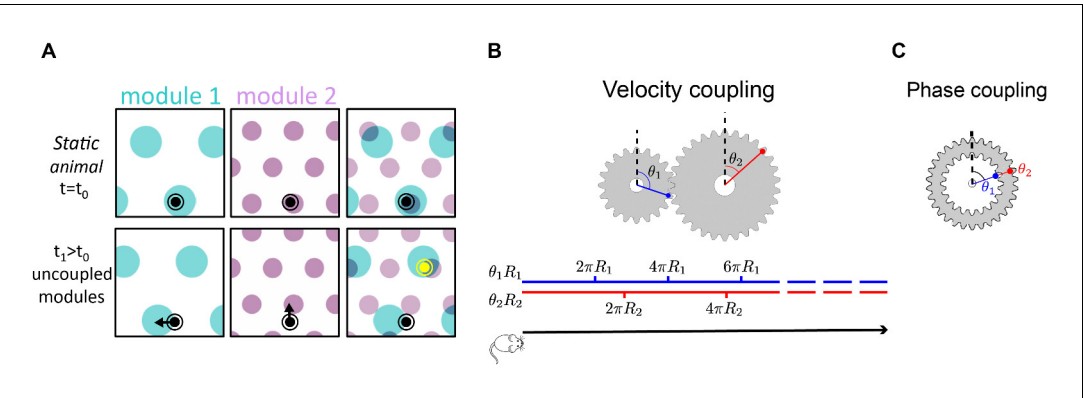

**Figure 1.** Velocity coupling: illu. (A) Illustration of the detrimental consequences arising from uncoupled module drifts. Black dot: location of a static animal. Top panels: schematic representation of the decoded position from the neural activity in module 1 (left panel) and module 2 (middle panel) at time $t = t_0$. The shaded areas (cyan, purple) represent locations whose likelihood, given the neural activity, is high. Top right panel: overlay of the likelihood read out from module 1 and module 2. The maximal likelihood location, based on activity in both modules, coincides with the animal's position. Bottom panels: decoded position based on the neural activity at time $t_1$. Due to independent, noise-driven drifts in each module, activity in module 1 represents positions that are slightly shifted to the left (bottom left), and activity in module 2 represents position that are slightly shifted upward (middle). Even though the shifts are small, the joint activity in both modules (bottom right) now represents a new maximum likelihood location (yellow), far away from the true location (black). We refer to such events as catastrophic readout errors. (B) Representation of position along a one-dimensional axis (black line) by the rotation angles $\theta_1$ and $\theta_2$ of two meshing gears that rotate in a coordinated manner in response to motion. The angles $\theta_1$ and $\theta_2$, corresponding to each position, are shown along the blue and red lines. If the radii $R_1$ and $R_2$ are incommensurate, a large range of positions can be unambiguously read out from the combination of the two angles. (C) An example of two phase coupled meshing gears that are fixed to each other, such that their angles $\theta_1$ and $\theta_2$ are always identical. Since $\theta_1 = \theta_2$, there is effectively only one encoded angle, and the range of unambiguous representation corresponds to a single rotation of the gears.

DOI: https://doi.org/10.7554/eLife.48494.002

Motivated by this analogy, we ask whether synaptic connectivity between grid cell modules can enforce a similar dynamic relationship between their states. Below, we identify a simple form of synaptic connectivity between grid cells that can implement this desired form of coupling. Next, we show that the recurrent connectivity confers the joint representation of position with resilience to two types of noise: First, noise in the velocity inputs projecting to different modules. These may differ in different modules, for example, due to synaptic noise. Second, noise arising from the stochastic dynamics of spiking neural activity within each module. The outcome is a continuous-attractor representation of position that achieves two goals: First, the representation is distributed across several modules with different spatial scales, allowing for combinatorial capacity by preserving the high dimensionality of the accessible steady states. Second, the neural circuitry that supports this representation is highly robust to noise when updating the representation based on self-motion, or while maintaining a persistent representation in short-term memory.

Alongside the recurrent connectivity, it is plausible that feedforward synaptic projections from the hippocampus to the entorhinal cortex play a role in shaping the grid cell response (*Kropff and Treves, 2008*; *Dordek et al., 2016*; *D'Albis and Kempter, 2017*; *Weber and Sprekeler, 2018*). Thus, hippocampal inputs may aid in coupling the states of different grid cell modules (*Welinder et al., 2008*; *Sreenivasan and Fiete, 2011*). In addition, sensory inputs that carry information about the animal's position may contribute as well, through synaptic projections to the MEC from other cortical areas. However, there are situations in which these types of inputs to the MEC cannot ensure appropriate coordination between grid cells modules. First, under conditions in which sensory inputs are absent or weak, the brain must rely on idiothetic path integration in order to update its estimate of position. Second, in novel environments, and following global remapping in the hippocampus (*Muller and Kubie, 1987*), it is highly unlikely that specific connections between place cells and grid cells, that couple the two spatial representations, are immediately established.

Hence, coupling modules via hippocampal inputs would be ineffective in a novel environment. Thus, in this study, we focus on the ability of recurrent connectivity *within* the entorhinal cortex to maintain a coherent representation of position across grid-cell modules.

## Results

### Theoretical framework

In laying out the principles underlying our proposed synaptic connectivity, we consider first a one dimensional analogue of the grid cell representation, inspired by the analogy to meshing gears discussed above: we imagine that an animal moves in one dimension, and neurons in each grid cell module $\mu$ jointly represent the modulus of position relative to the grid spacing $\lambda_\mu$ (for simplicity, from here on, we define the phases $\theta_\mu$ such that they are in the range $[0, 1]$):

$$\theta_\mu = \frac{x \bmod \lambda_\mu}{\lambda_\mu}. \tag{1}$$

We hypothesize that the joint dynamics of all grid cells within a module are restricted to lie in a one dimensional attractor, which we model as a ring attractor (*Ben-Yishai et al., 1995*; *Zhang, 1996*). More specifically, we consider the double-ring architecture (*Xie et al., 2002*), which includes a mechanism for updates based on velocity inputs, and was proposed as a model for integration of angular velocity inputs by head direction cells in rodents. Recent discoveries in the *Drosophila melanogaster* central complex point to a representation of heading that is maintained by neural circuitry with close similarity to this architecture (*Seelig and Jayaraman, 2015*; *Turner-Evans et al., 2017*; *Kim et al., 2017*; *Green et al., 2017*). Attractor models of grid cells in the entorhinal cortex (*Fuhs and Touretzky, 2006*; *Burak and Fiete, 2009*) generalize the double-ring attractor model to motion in two dimensions.

Within the double-ring attractor model (*Xie et al., 2002*), a module consists of two recurrently connected neural sub-populations, each comprising $N$ neurons organized on a ring (left ring and right ring, *Figure 2A*). We denote by

$$\vec{s} = \begin{pmatrix} \vec{s}_R \\ \vec{s}_L \end{pmatrix} \tag{2}$$

the vector of synaptic activations, where $\vec{s}_R$ and $\vec{s}_L$ represent the synaptic activation of the right and left sub-populations respectively. The synaptic activation $s_i$ of neuron $i$ follows the dynamics:

$$\dot{s}_i + \frac{s_i}{\tau} = r_i = \phi\left(\sum_{j=1}^{2N} W_{ij}s_j + I_0 \pm dI\right), \tag{3}$$

where $\tau$ is the synaptic time constant, $r_i$ is the firing rate of neuron $i$, $\phi$ is a nonlinear transfer function, $W$ is the connectivity matrix (*Equation 8*; *Equation 9*), $I_0$ is a constant input, and $+dI$ ($-dI$) is the velocity input to a neuron in the right (left) sub-population. Synaptic weights projecting from neurons in each ring are shifted clockwise (right) or anti-clockwise (left). When both sub-populations receive identical feed-forward inputs, activity in the network settles on a stationary bump of activity. However, selective activation of the right (left) sub-population via the feed-forward inputs, induces clockwise (anti-clockwise) motion of the activity bump at a phase velocity proportional to the velocity input $dI$ (*Figure 2B*). Hence, in a noise-free network, the position of the activity bump is an integral of the velocity input.

Our goal is to couple several such modules such that they will update their states in a coordinated manner in the presence of noisy inputs. It is essential to couple the modules based on their phase velocities $\dot{\theta}_\mu$ and not directly by their phases $\theta_\mu$, as we want to allow all phase combinations of the different modules to be possible steady states of the population neural dynamics. Our proposed coupling requires two ingredients: reading out the phase velocity of each module, and inducing corresponding phase velocities in the other modules. The double ring model already contains a mechanism for integration of velocity inputs, and therefore, our main challenge is in reading out the phases velocities.

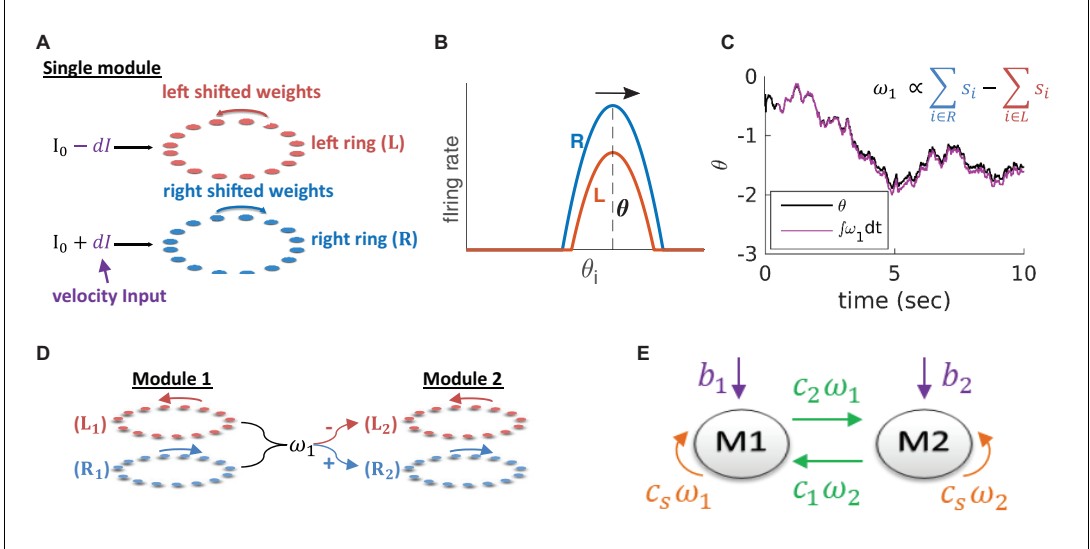

**Figure 2.** Model architecture. (A) Structure of a single module, consisting of a left ring (red) and a right ring (blue), in accordance with the double ring model (*Xie et al., 2002*). The two rings receive external inputs proportional to velocity, with opposite polarities (purple). Synaptic weights project slightly anti-clockwise (red) and slightly clockwise (blue) from neurons in the left and right sub-populations. (B) Illustration of the firing rates of the right (blue) and left (red) sub-population during velocity integration. Both activity bumps are centered around the same phase. The right population is receiving stronger feed-forward input than the the left population, due to a positive velocity signal (*dI* in (A) and *Equation 3*). Since the outgoing synaptic weights of the right sub-population project clockwise, the activity bump of both populations moves to the right. (C) True phase $\theta$ as a function of time (black) in response to an external velocity input, representing a simulated trajectory, and an estimation of this phase from the velocity approximation $\omega_1$ (magenta). Note that $\theta$ is periodic with period 1, but for presentation clarity we unwrap $\theta$ to depict a continuous path along the real axis. (D) The coupling of drifts in two modules is achieved by providing the velocity approximation $\omega_1$ as a velocity input to module 2 (and vice versa, not shown). Each module is modelled as a double ring attractor, as in (A). (E) Two coupled modules. The velocity input of each module has three contributions: The external velocity input (purple), the coupling of velocity from the other module (green), and the self coupling (orange).

DOI: https://doi.org/10.7554/eLife.48494.003

## Simple neural readout of velocity

Our first goal is to read out the phase velocity of a single module in our system. It is possible to compute the phase velocity by projecting *Equation 3* on the eigenvector with zero eigenvalue of the dynamics (see Appendix 1 and *Burak and Fiete, 2012*). However, this projection cannot be evaluated linearly from the neural activity, since the projection coefficients depend on the location of the activity bump. Instead, we seek a simple estimate of the phase velocity that can be implemented in a neural circuit with relatively simple architecture in a biologically plausible manner.

Intuitively, in the described framework, most of the motion arises from the differences in activity between the right and left sub-populations *Figure 2B*. Therefore, this difference might be close to the phase velocity $\dot\theta$. We find, indeed, that the difference between the synaptic activities of the right and left sub-populations,

$$\omega \equiv \frac{\beta}{\tau}\left(\sum_{i\in R}s_i - \sum_{i\in L}s_i\right) \tag{4}$$

provides a good approximation for the phase velocity (*Figure 2C*), where $\beta$ is a proportionality factor. In Appendix 1 we show mathematically that $\omega\approx\dot\theta$.

## Coupling modules by synaptic connectivity

In order to couple the motion of different modules, we use the readout signal $\omega_\mu$ of each module $\mu$ (*Equation 4*) as a velocity input to all other modules (*Figure 2D–E*, green arrows). In addition, we include negative self coupling within each module using the same readout signal $\omega_\mu$ (necessary, as shown below, in order to prevent instabilities that otherwise arise from the positive feedback generated by the inter-module couplings), *Figure 2E* (orange arrows).

Note that $\omega_\mu$ is a linear function of synaptic activities within the ring network, with coefficients that do not depend on the position of the activity bump. Thus, the coupling can be implemented by recurrent connectivity within the MEC, between modules and within a single module. The resulting synaptic connectivity between any two coupled modules is all-to-all in the sense that every neuron in one module is connected to every neuron in the other module, with synaptic weights whose magnitudes are uniform (see Materials and methods, *Equation 12*). The sign of each synaptic weight depends only on the sub-population (left or right) of the pre- and post-synaptic neurons. This connectivity is completely symmetric to rotation in the two modules, thus preserving the ability to obtain a combinatorially large manifold of steady states in which activity bumps can be placed in any combination of positions.

To understand how the couplings influence the joint dynamics of the coupled modules, we analyze the response of a network, consisting of $m$ coupled modules, to external velocity inputs, $\vec{b}(t)$. The position of the bump in each module can be represented by a phase $\theta_\mu$. We find that the dynamics of these phases are governed by the following set of coupled differential equations

$$\dot{\vec{\theta}} = \alpha\vec{b}(t) + C\left(f * \dot{\vec{\theta}}\right), \tag{5}$$

where $C$ is an $m \times m$ matrix whose element $C_{\mu\rho}$ represents the coupling strength from module $\rho$ to module $\mu$, $f * \dot{\vec{\theta}}$ is the convolution of $\dot{\vec{\theta}}$ with an exponential filter $f$ with the synaptic time scale $\tau$ (*Equation 36*), and $\alpha$ is a constant factor (see full derivation of *Equation 5* in Appendix 2). Thus, the phase of each module is updated in response to two signals: the external velocity input projecting into the module (first term in *Equation 5*), and the recent history of changes in the phases of the other modules, conveyed by the coupling signal (second term in *Equation 5*).

Much of the system's response to external velocity inputs can be understood by considering its dynamics under the assumption that the motion of the animal is sufficiently slow, such that the components of $\dot{\vec{\theta}}$ vary slowly compared to the synaptic time constant. Under this assumption, we obtain from *Equation 5*

$$\dot{\vec{\theta}} = X \cdot \vec{b}(t), \tag{6}$$

where

$$X \equiv \alpha(I - C)^{-1} \tag{7}$$

is the *linear response tensor*.

For simplicity, let us consider first only two coupled modules (each of them one dimensional), with identical self coupling strength $C_s$ for both modules. The eigenvalues of $X$, denoted by $X_+$ and $X_-$ (*Figure 3A–C* and Appendix 2), indicate how strongly the coupled modules respond to velocity inputs that drive coordinated and relative motion, respectively. If $X_-$ is small, the modules respond weakly to velocity inputs that attempt to update the phases in an uncoordinated manner. Thus, if $X_-$ is much smaller than $X_+$, we expect the motion of the two modules to remain coordinated, even if the velocity inputs to the two modules differ.

We choose coupling parameters $C_1$, $C_2$ and $C_s$ such that three requirements are fulfilled (see Appendix 2): First, the modules should respond significantly to inputs that drive coordinated motion (large $X_+$, *Figure 3A,C*). The response to such inputs should not be suppressed since the system must be able to update its state based on velocity inputs, to correctly represent the animal's position in its environment. Second, the modules should respond very weakly to inputs that drive anti-correlated motion (small relative motion $X_-$, *Figure 3B,C*). The self negative coupling $C_s$ enables us to achieve these two requirements while preserving stability (*Figure 3A–C* and Appendices 2-3). Our last requirement is the maintenance of a specific ratio between the module phase velocities, $\lambda$, that corresponds to the grid spacing ratio between successive modules (we set $\lambda = \sqrt{2}$ for all modules; *Stensola et al., 2012*).

*Figure 3E–F* demonstrates the response of two modules to an external velocity input, representing an animal's trajectory (shown in *Figure 3D*). The input is given only to module 1. In the the case of uncoupled modules ($C_1 = C_2 = C_s = 0$), only module 1 follows the trajectory, as expected (*Figure 3E*). In the case of coupled modules, both of the modules follow the trajectory quite

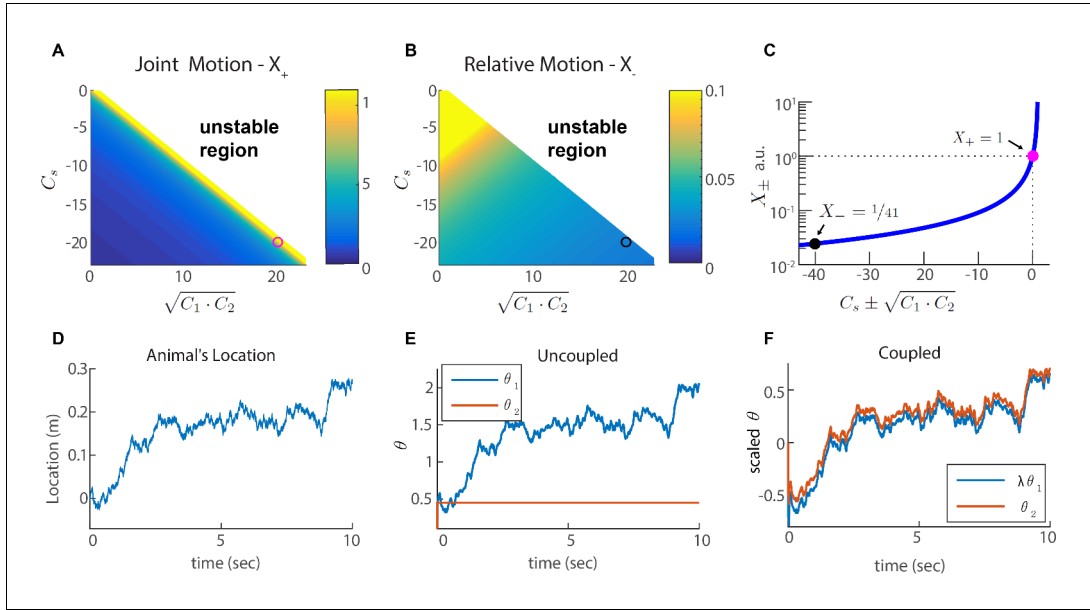

**Figure 3.** Two coupled modules. (**A-B**) Velocity response of the coupled system to velocity inputs that drive joint motion $X_+$. (**A**) or relative motion $X_-$(**B**) in two coupled modules, as a function of the coupling strengths. (**C**) $X_\pm$ is a function of one parameter that depends on the coupling strengths ($C_s \pm \sqrt{C_1 C_2}$). The magenta and black circles represent the parameters used in (**F**), and the corresponding value of $X_+$ and $X_-$. (**D**) Simulated trajectory, whose derivative is injected as a velocity input only to module one in panels (**E–F**). (**E**) Response of two uncoupled modules ($C_1 = C_2 = C_s = 0$): the position represented by module 1 tracks the velocity inputs, module 2 is unresponsive, and the updates in the two modules are not coordinated, as expected. (**F**) Same as (**E**) but the modules are coupled with coupling strengths: $C_s = -20$, $C_1 = -C_s/\lambda \simeq 14.1$, and $C_2 = -C_s \cdot \lambda \simeq 28.3$. The phases of both modules track the velocity inputs in a coordinated manner, with the desirable velocity ratio $\lambda$. The phase of module one is scaled by $\lambda$ ($\lambda\theta_1$ is shown) in order to simplify the comparison between modules.
DOI: https://doi.org/10.7554/eLife.48494.004

accurately, with the desired phase velocity ratio $\lambda$. Hence, under these conditions, the two coupled modules shift in a coordinated manner, even if they receive incompatible velocity inputs. Next, we generalize these results to multiple modules, and to grid cells in two dimensions.

## Generalization to two dimensions and several modules

The coupling of modules, described so far, can be easily extended to grid cell responses in two dimensions. In accordance with grid cell responses in two-dimensional arenas, each module is structured as a two-dimensional attractor, whose state is determined by two periodic phases. Thus, the steady states of the attractor are arranged on a torus instead of a ring. To obtain this topology, we use a network architecture in which neurons are arranged on a parallelogram, corresponding to a unit cell of the hexagonal grid (in similarity to *Guanella et al., 2007*). The synaptic connection between any two neurons depends on their distance on the parallelogram, defined using periodic boundary conditions (*Figure 4A* and Materials and methods). The position of the activity bump must be able to shift in response to a two-dimensional velocity input, in any direction in the plane. To implement this velocity response, each module contains four sub-populations (right, left, up, and down). The synaptic weights projecting from neurons in each sub-population are shifted in a corresponding direction within the neural population (*Burak and Fiete, 2009*).

In addition, we generalize our network to $m$ grid cell modules. The coupling strengths $C_{\mu\rho}$ thus comprise $m^2$ parameters that we are free to adjust to fulfill a set of requirements, similar to those applied in the case $m = 2$. Our most important goal is that the motion of all modules should be coordinated, even if the velocity inputs are not identical. To achieve this goal, we define a joint motion vector $\vec{u}$, such that $u_\mu/u_\rho$ is the ratio of grid spacings of modules $\rho$ and $\mu$. We require that this vector is an eigenvector of the linear response tensor, and minimize the eigenvalues corresponding to all

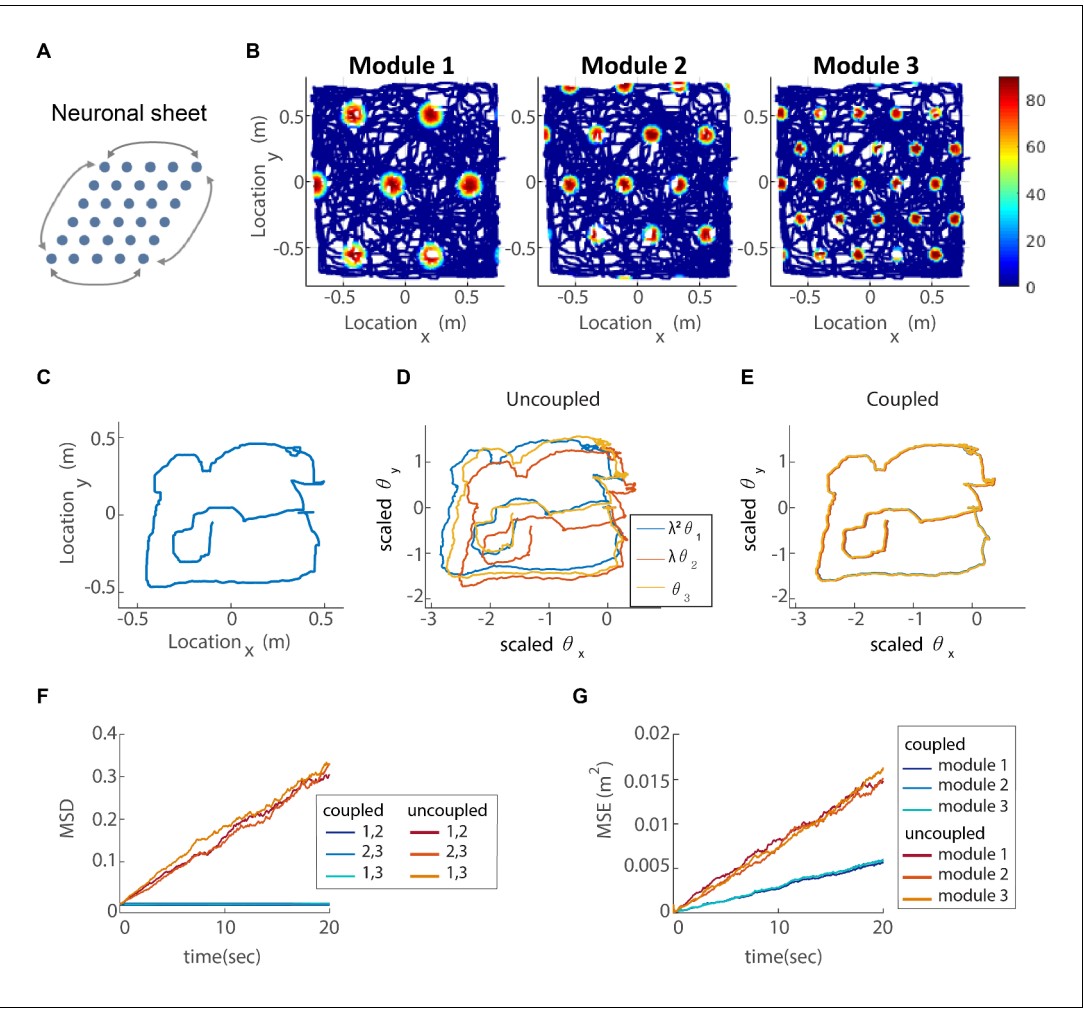

**Figure 4.** Coupling of several modules in two dimensions. (**A**) The neurons of each sub-population in the two dimensional case (right, left, up and down) are organized on a neuronal sheet in the shape of a parallelogram with periodic boundary conditions. (**B**) Simulated firing rate of a single grid cell from each module, as a function of position, evaluated during response of the network to a rat trajectory lasting 800 s (taken from *Stensola et al., 2012*). (**C**) Measured rat trajectory over an interval of 20 s (*Stensola et al., 2012*), whose derivative is injected as a velocity input to all modules in panels (**D–E**), with addition of uncorrelated noise in each module. (**D**) Response of three uncoupled modules. (**E**) Response of three coupled modules. The phases of the three modules approximately track the velocity inputs in both cases, but the coordination between phases of the three modules is more tight in (**E**). The phases of modules 1 and 2 are scaled by $\lambda^2$ and $\lambda$, respectively, in order to simplify the comparison between modules (similar to *Figure 3F*). (**F**) Mean square displacement (MSD) between the scaled phases of any two modules over time. Responses of the three modules, as in (**D–E**), were simulated over a hundred realizations of noise. The mismatch between module trajectories in the coupled case (blue lines) is very small compare to the uncoupled case (red lines). The units of scaled phases are the same as in (**D–E**). The legend indicates which two module trajectories are compared. (**G**) Mean square error (MSE) of each module's trajectory relative to the animal's trajectory, computed from the same simulations presented in (**F**). To obtain each module's trajectory in units of spatial location, the module's phase was multiplied by its spacing. In the coupled case (blue), the slope of MSE is reduced by a factor of $m = 3$ (the number of modules) compared to the uncoupled case (red), as the noise is averaged due to the coupling.

DOI: https://doi.org/10.7554/eLife.48494.005

The following figure supplement is available for figure 4:

**Figure supplement 1.** Disconnected module.

DOI: https://doi.org/10.7554/eLife.48494.006

other eigenvectors. If we were able to obtain eigenvalues that precisely vanish, the response tensor would be a rank one matrix whose columns are all proportional to $\vec{u}$. Under this idealized outcome, any velocity input, regardless of its direction in the $2m$ dimensional input space would result in coordinated motion of the modules. However, the couplings $C_{\mu\rho}$ that precisely achieve this goal diverge, in similarity to the two-module case (Appendix 2). Thus, we impose a constraint on the strength of the synaptic connections. Another constraint is that all eigenvalues of $C$ must be smaller than unity. Otherwise, the system exhibits dynamic instability (Appendix 3). We optimize an appropriate target function under these constraints (Appendix 3).

For $m = 2$ the optimization results in the same solution of coupling parameters $C_{\mu\rho}$ that we found previously. For $m > 2$, we find that there is considerable freedom in choosing combinations of $C_{\mu\rho}$ that achieve satisfactory coupling (See Appendix 3). One principled way to reduce this freedom, is to require that there is connectivity only between successive modules. This choice is compatible with recent observations (*Fu et al., 2018*) that excitatory synaptic connectivity within the MEC is relatively short ranged. In our numerical results, we use this assumption to constrain the structure of the connectivity matrix $C_{\mu\rho}$, but other choices that include broader connectivity between modules lead to similar coupling between the modules.

To demonstrate how our proposed coupling affects the response of the modules to velocity inputs, we simulate the described network in two dimensions, with and without coupling, and with three modules. The velocity input is a measured rat trajectory from *Stensola et al. (2012)*, with the addition of white Gaussian noise, drawn independently in the three modules. In each module, we set the proportionality coefficient that tunes the modulation of activity by the velocity input ($\gamma_\mu$ in *Equation 21*) to achieve the desired grid spacing, even in the absence of inter-module coupling. In the simulation, we assume that only velocity inputs are responsible for the update of the neural representation of position, thus mimicking a situation in which sensory cues, such as those arising from visual inputs and encounters with the walls (*Hardcastle et al., 2015*; *Keinath et al., 2018*), are absent. In a noise-free simulation, the single cell firing rates form a hexagonal grid pattern as a function of the animal's location (*Figure 4B*), as expected from the network structure, while the spacing ratio between modules is close to $\lambda$.

The trajectories of the 2d phases, in response to noisy velocity inputs, are shown, for each of the three modules, in *Figure 4D* (uncoupled modules) and *Figure 4E* (coupled modules). In both cases, the phases follow the animal's trajectory (*Figure 4C*) quite closely, but the phases are much more similar to each other, and to the original trajectory, in the coupled case. Since panels D-E show results only from a single simulation, we repeat the analysis for 100 realizations of the noise in the velocity inputs, to obtain statistical measures on the coupled vs. uncoupled dynamics. The coupling substantially reduces the mismatch accrued between the trajectories of the different modules, compared to the uncoupled case, *Figure 4F*. For comparison, *Figure 4G* shows the mismatch between module trajectories and the true trajectory. In the uncoupled case, all modules exhibit similar accumulation of error, which arises from their independent responses to the noise in the velocity inputs. In the coupled case, only the projection of the noise on the direction of joint motion contributes to the accumulation of errors, leading to a reduction by a factor of $m$ (in our case 3) in the slope of the MSE curve.

The lack of deviations between the phase trajectories, seen in the coupled case (*Figure 4E–F*), is an essential difference between the dynamics of the coupled and uncoupled modules. As discussed in the *Introduction*, we expect this difference to strongly impact the stability of the grid cell code. In the following section, we substantiate this point.

## Consequences for spatial representation and readout

We next aim to validate our hypothesis that the coupling of modules stabilizes the grid cell code, and more specifically, prevents catastrophic errors that can be caused by uncoupled drift in the phases of different modules (*Figure 1A*). We simulate the dynamics of our three module network with noisy velocity inputs based on a measured rat trajectory from *Stensola et al. (2012)*, as in *Figure 4*. We then generate Poisson spikes from the instantaneous firing rates of the neurons, and read out the animal's trajectory from the simulated spikes: we do so both for coupled modules (*Figure 5A*) and for uncoupled modules (*Figure 5B*). The readout is accomplished using a decoder that sums spikes from the recent history, with an exponentially decaying temporal kernel (see

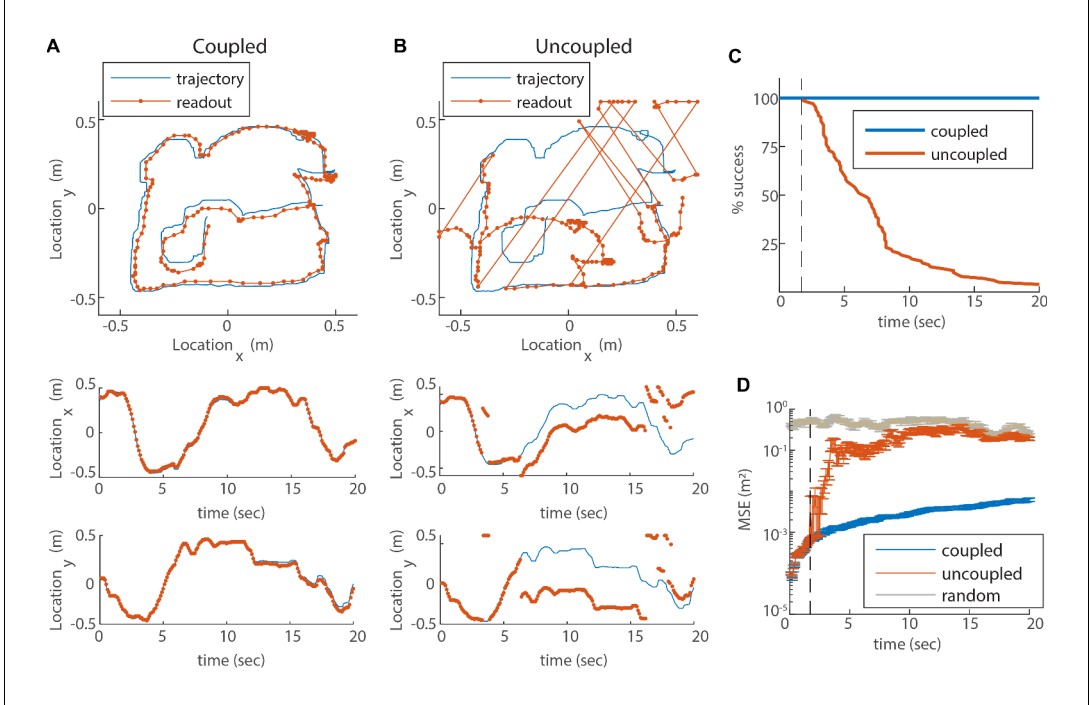

**Figure 5.** Resilience of the spatial representation to noise in velocity inputs. (**A**) Blue trace: measured rat trajectory over a 20 s interval (taken from *Stensola et al., 2012*), used as a velocity input to three coupled modules. Red trace: readout of position, decoded from simulated Poisson spikes of the coupled system. The spikes are generated by a Poisson process from the instantaneous firing rates of all cells in the three modules (see Materials and methods). Top panel: the trajectory in the two-dimensional arena. Lower panels: *x* and *y* components of the trajectory as a function of time. The decoded position is continuous and similar to the input trajectory. (**B**) Same as (**A**), but in a network consisting of three uncoupled modules: all coupling strengths are set to zero. The decoded position is discontinuous in time, and often sharply deviates from the input trajectory. (**C**) Percentage of decoding success over time. We repeated the decoding of the animal's trajectory, as in (**A–B**), over a hundred simulations with different realizations of the noise. A success at time *t* is defined as a trial that did not contain any discontinuity in the readout up to that time. The success percentage was computed by counting the number of trials without discontinuities at each time point. The coupled network maintains 100% success over time (blue), whereas the success percentage of the uncoupled network decreases significantly over time: many trials contain discontinuities after a few seconds, and almost all of them contain discontinuities after 20 s (red). (**D**) Mean square error (MSE) of the decoded trajectory, computed from the same simulations presented in (**C**), in the coupled (blue) and uncoupled (red) cases. Gray trace: MSE computed by random guessing of location. The vertical black dashed line in (**C–D**) represents the first time in which a discontinuity was observed in any of the trials of the uncoupled network. From this time point onward, the percentage of success of the uncoupled network descends, and the MSE sharply increases (note the logarithmic vertical scale).

DOI: https://doi.org/10.7554/eLife.48494.007

The following figure supplements are available for figure 5:

**Figure supplement 1.** Effect of the readout integration time scale.

DOI: https://doi.org/10.7554/eLife.48494.008

**Figure supplement 2.** Effect of the number of modules and environment size.

DOI: https://doi.org/10.7554/eLife.48494.009

Materials and methods and *Mosheiff et al., 2017*). In *Figure 5* the spikes are involved only in the readout process and not in the intrinsic dynamics of the neural network.

In the case of coupled modules the decoded trajectory is similar to the input (*Figure 5A*), but due to the noise in the inputs, it gradually accrues an error relative to the true trajectory. Without coupling, the position read out from the network activity diverges sharply from the true trajectory (*Figure 5B*). Moreover, the readout trajectory is often discontinuous in time, and thus cannot be a good approximation to any reasonable path of the animal. The discontinuity arises from uncorrelated drifts of the modules which, combined with the periodic nature of the grid pattern, cause catastrophic readout errors, much larger than the errors accrued in the phases of each module separately (*Figure 1A*).

In order to quantitatively substantiate the relationship between the large deviation of the decoded trajectory from the true trajectory and the occurrence of catastrophic readout errors, we

repeat the decoding process a hundred times with and without coupling, for a 20s input trajectory. In all realizations with coupling, the readout is coordinated with the input trajectory (*Figure 5C*, blue). In contrast, without coupling almost all realizations exhibit discontinuities within a time interval of 20s (*Figure 5C*, red). The mean square error (MSE) of the decoder increases as a function of time in the coupled as well as the uncoupled systems (*Figure 5D*), as expected due to noise in the input (as the coupling and inputs are of velocity and not location, there is no correcting mechanism that can correct coordinated shifts in the phases of all the modules). However, a few seconds after the start of the simulation, the MSE grows sharply in the uncoupled system (dashed line in *Figure 5D*). The time at which this starts to happen coincides with the first appearance of discontinuities in the decoded position (compare *Figure 5* panels C and D). (Note that below this time the probability for occurrence of a readout discontinuity does not vanish, but can be inferred roughly to be small compared to 0.01 since we performed 100 simulations.) Thus, the dramatic reduction achieved by the coupling between modules arises primarily from the elimination of catastrophic readout errors. This conclusion is insensitive to the choice of the time scale of temporal integration used in the decoding process (*Figure 5—figure supplement 1*).

Qualitatively similar conclusions, as demonstrated above, are obtained also when the number of modules $m$ is increased (*Figure 5—figure supplement 2*). With larger $m$, the rate at which readout discontinuities occur in a given environment diminishes. Note, however, that additional modules enable unambiguous representation of larger environments (*Fiete et al., 2008*; *Mathis et al., 2012a*; *Wei et al., 2015*; *Vágó and Ujfalussy, 2018*), and that the rate of readout discontinuities increases with the size of the environment (red traces in *Figure 5—figure supplement 2E–J*).

## Intrinsic neural noise

Up to this point we presented a theory of several grid cell modules, coupled to each other by synaptic connectivity within the MEC, such that the coupling significantly suppresses incompatible drifts caused by noisy inputs to the system. Next, we wish to address another important source of noise, arising from the variability in the spiking of individual neurons within the grid cell network (*Softky and Koch, 1993*; *Shadlen and Newsome, 1994*; *Burak and Fiete, 2009*). In similarity to noise in the inputs, stochasticity of the neurons participating in the attractor network drives errors that accumulate over time with diffusive dynamics (*Burak and Fiete, 2009*; *Burak and Fiete, 2012*). To model this process, we replace the firing rate of each neuron in *Equation 3* by a Poisson spike train (see *Equation 22*).

Since we designed our network to be resilient to noisy inputs, it is not obvious that the same architecture can also provide resilience to intrinsic noise. To address this question, we revisit first the simple case of two coupled modules in one dimension. In Appendix A.2 we show that the simple readout of velocity used to couple the modules (*Equation 4*), is a good approximation for the phase velocity driven by intrinsic neural noise, suggesting that the coupling introduced previously can help suppress uncoordinated drifts. To quantify the impact of coupling on coordination of the modules, we compute the diffusion tensor of their phases, using *Equation 25* (the calculation is based on the theoretical framework laid out in *Burak and Fiete, 2012*; see specifically Eq. S24). In the uncoupled case, the diffusion tensor is isotropic as expected (*Figure 6A*, blue line). When the modules are coupled, with the same coupling strengths as in *Figure 3*, the diffusion of the two modules is highly anisotropic (*Figure 6A*). The first principal axis of the diffusion tensor (red ellipse in *Figure 6A*) closely matches the direction of coordinated motion (dashed line in *Figure 6A*). The diffusion coefficient $D_-$, associated with motion in the orthogonal direction, is much smaller than the diffusion coefficient $D_+$, associated with coordinated motion: $D_+/D_- \sim X_+/X_- \sim 40$ (compare Figures *Figure 3C* and *Figure 6A*). Thus, the coupling strongly suppresses incompatible diffusion of the two modules.

Next, we evaluate the consequences for representation and readout, arising from the suppression of incompatible diffusion arising from intrinsic neural noise. We repeat the simulation of three coupled modules in two dimensions, this time with stochastic (Poisson) neurons. Discontinuities in the decoded trajectory occur both in the uncoupled and coupled networks, but they are much more rare in the coupled network (*Figure 6B*). Accordingly, the readout MSE is reduced dramatically in the coupled network (*Figure 6C*, note the logarithmic scale). Thus, the coupling is effective not only in stabilizing the neural representation in response to noisy inputs, but also with respect to internal stochasticity within the grid cell network.

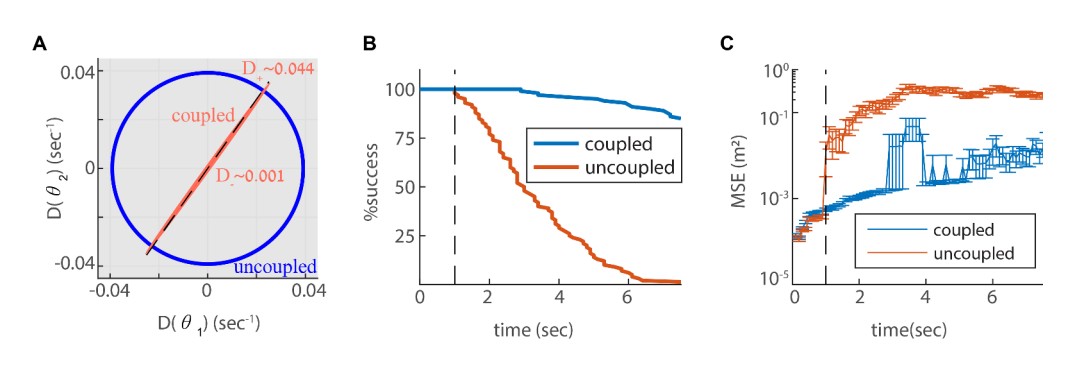

**Figure 6.** Reilience of the spatial representation to intrinsic neural stochasticity. (**A**) Diffusion tensor of a Poisson spiking neural network, consisting of two modules in one dimension, computed using *Equation 25*, and illustrated as an ellipse. Axes of the ellipse are aligned with the eigenvectors of the diffusion tensor, and the lengths of each axis represents the diffusion coefficient along the corresponding direction. Without coupling the diffusion tensor is isotropic (blue circle). When coupling the modules using the same coupling strengths as in *Figure 3F*, the diffusion tensor becomes highly anisotropic (red ellipse). The diffusion in this case is almost exclusively in the direction of the first principal component (major axis of the ellipse). This direction closely matches the direction of coordinated drift ($\bar{u}_+$ in *Equation 55*, dashed line). (**B-C**) same as *Figure 5C–D*, but for the internal noise case.

DOI: https://doi.org/10.7554/eLife.48494.010

In principle, one could seek coupling parameters such that diffusion would be suppressed in all directions. However, recall our first requirement from the section *Coupling modules by synaptic connectivity*, that the network must respond with sufficient gain to external inputs, to follow an animal's trajectory. As we keep the joint response strong, we cannot reduce the joint diffusion simultaneously, and we are satisfied with coupling of the diffusive drift, without eliminating coordinated diffusion.

## Discussion

Previous works (*Fiete et al., 2008*; *Mathis et al., 2012b*; *Wei et al., 2015*; *Mosheiff et al., 2017*) have shown that grid cell activity, viewed as a neural code for position, achieves a high dynamic range due to the splitting of the representation across multiple modules. In this work, we addressed a key difficulty with this idea: the combinatorial nature of the representation, arising from the existence of multiple modules, leads to high vulnerability to noise. Small uncoordinated errors in the phases of the different modules can shift the represented position to a far away location. As a possible solution to this difficulty, we proposed a simple architecture of synaptic connectivity between grid cell modules, that can suppress incompatible drifts. The functional coupling between modules, arising from our proposed synaptic connectivity involves velocities, but is completely insensitive to their phases. Consequently, the coupling does not limit the combinations of possible phases of the different modules, and thus does not affect the capacity of the code.

Similar principles may apply to storage in working memory and coding of other continuous, low dimensional variables in the brain. Thus, the main contribution of our work from the theoretical perspective, is that it identifies a way to couple several low dimensional continuous attractors of dimension $d$ (in the case of grid cells, $d = 2$), to produce a persistent neural representation of a single, $d$ dimensional variable with high dynamic range. The dynamics of the network are characterized by two seemingly contradictory features: first, the steady states of the system span a space of dimension $md$, where $m$ is the number of modules. Second, during maintenance and continuous update of the stored memory, the joint state of the modules is dynamically restricted to lie in a much smaller, $d$-dimensional subspace. This enables the continuous embedding of a $d$ dimensional variable in the larger, $md$ dimensional space, without allowing for noise to shift the state of the system outside the appropriate, $d$ dimensional local subspace.

In the simulations of the coupled and uncoupled grid cell networks (*Figure 5*; *Figure 6*), our main goal was to demonstrate that with a reasonable choice of parameters, catastrophic readout errors are highly detrimental, and that the coupling mechanism greatly reduces the rate at which they

occur. The rate of catastrophic readout errors, quantified in *Figure 5C* for specific choices of parameters, depends also on the noise sources, and on the probability that a set of shifted phases might match an alternative position in the environment. The latter quantity is influenced by the size of the environment, the number of modules, and the specific grid spacings and orientations (*Fiete et al., 2008*; *Welinder et al., 2008*; *Burak and Fiete, 2009*; *Sreenivasan and Fiete, 2011*; *Burak, 2014*; *Vágó and Ujfalussy, 2018*) (see also *Figure 5—figure supplement 2*).

The number of grid cell modules in the entorhinal cortices of rats and mice is unknown. So far there is direct evidence for the existence of four modules, but the number may be larger (*Stensola et al., 2012*; *Rowland et al., 2016*). A theoretical attempt to compare grid cell systems with different number of modules in terms of the rate of readout discontinuities, would require additional assumptions on the range of positions that are represented in each system: increasing the number of modules reduces the rate of readout discontinuities within the range of a given environment, but it offers the possibility to unambiguously represent larger environments (*Fiete et al., 2008*; *Mathis et al., 2012a*; *Wei et al., 2015*; *Vágó and Ujfalussy, 2018*), for which the error rate is higher *Figure 5—figure supplement 2E–J*. Furthermore, the gain in capacity of the grid cell code, obtained with addition of modules, may be harnessed by the entorhinal-hippocampal system to generate unique representations of different environments (*Fyhn et al., 2007*). Thus, incoherent phase errors may lead to confusion between different spatial maps, in addition to the confusion between two positions in any given environment. Accordingly, the rate of catastrophic readout errors may be influenced by the number of spatial maps represented in the brain.

Our proposed mechanism for coupling modules is complementary to another possible mechanism, of coupling grid cell modules through the reciprocal synaptic connectivity between the entorhinal cortex and the hippocampus (*Welinder et al., 2008*; *Sreenivasan and Fiete, 2011*; *Burak, 2014*). Since biological systems often harness multiple mechanisms to achieve the same function, both mechanisms might act in parallel to stabilize the grid cell code against catastrophic readout errors. As discussed in the introduction, it is highly unlikely that coupling via the hippocampus could work in a novel environment, following global remapping. On the other hand, it is of particular importance for the brain to establish a geometric representation of position, aided by idiothetic path integration, under this scenario. Thus, the velocity coupling mechanism proposed in this work may play an especially important role in generating a cognitive map of a novel environment.

Inputs from cells within the MEC may play a role in stabilizing the grid cell representation, alongside inputs from the hippocampus or other areas. These may include inputs to grid cells from border cells (*Solstad et al., 2008*) or object-vector cells (*Høydal et al., 2019*). Experimentally, it has been demonstrated that phase resets occur in the grid cell representation upon encounters with enviornmental boundaries (*Hardcastle et al., 2015*; *Keinath et al., 2018*; *Ocko et al., 2018*), and it has been argued theoretically that such resets can be implemented in attractor models of grid cells by inputs to grid cells from border cells (*Hardcastle et al., 2015*; *Keinath et al., 2018*; *Ocko et al., 2018*; *Pollock et al., 2018*). The origin of spatial specificity of border cells and object-vector cells is not yet identified, but since both types of cells are active even when an animal is not facing the features associated with their activation, their role in stabilizing the grid cell representation may be similar to the hypothesized role of place cell inputs in stabilizing the grid cell code, perhaps more so than the role of direct sensory inputs.

A model that involves synaptic coupling between modules, of a different architecture than the one considered here, has been recently proposed in *Kang and Balasubramanian (2019)*. This model does not explore the consequences of noise on coding stability, and its primary goal is to explain the ratios between grid spacings, and the emergence of modularity (see also *Urdapilleta et al., 2017*). Hence, *Kang and Balasubramanian (2019)* address different questions from those studied in the present work. Nevertheless, it is plausible that the synaptic connectivity proposed by Kang and Balasubramanian stabilizes the dynamics against incompatible motion of the modules. An important difference between the network architecture explored in *Kang and Balasubramanian (2019)* and the architecture explored here, is that we consider connectivity between modules which is all-to-all (every grid cell in one module projects to every grid cell in the other module), and is designed to be invariant to any static, relative shift in the module phases. Hence, all combinations of phases are steady states of the dynamics. In contrast, the synaptic connectivity considered in *Kang and Balasubramanian (2019)* is spatially local. Consequently, it tends to produce interlocked patterns of activity in adjacent modules, with shared spatial periodicity, and preferred relative spatial phases. These

properties of the activity patterns are expected to limit the representational capacity of the code. Here, we addressed a different computational goal, of stabilizing a distributed representation of position over multiple modules, without compromising the dynamic range of the neural coding scheme.

Our focus in this work was on the suppression of relative motion across modules, but noise in the inputs, or in the intrinsic activity within the network, drives also coordinated motion. It is possible to suppress the latter type of random motion by increasing the negative feedback in the system (*Figure 3A,C*). In choosing our optimization goal for the coupling parameters, we did not attempt to suppress coordinated drift for two reasons. First, coordinated drift is much less detrimental from the coding perspective than relative drifts, as discussed in the introduction. Second, suppressing the coordinated motion comes with an inevitable cost: a reduction in the gain of the system's velocity response. Nevertheless, it is interesting to consider also the suppression of coordinated drift. Next, we briefly discuss the possible implementation of this goal.

For simplicity, consider a single one-dimensional module, structured as a ring attractor: in this situation, there is only coordinated motion. As in any continuous attractor network, stochasticity of neural activity within the network drives diffusive motion of the bump's position. This diffusive motion gradually degrades the fidelity of the stored memory (*Burak and Fiete, 2012*). In the double ring architecture (*Xie et al., 2002*), much of the drift arises from fluctuations in the difference of activity between the two sub-populations that drive left and right motion. Using the negative self-coupling of velocities, introduced in this work, it is possible to suppress these fluctuations to substantially reduce the noise-driven diffusion and stabilize the represented memory. It is interesting to compare this mechanism with another proposal (*Lim and Goldman, 2014*) for stabilization of the memory stored in a single ring attractor, using negative derivative feedback (*Lim and Goldman, 2013*). In (*Lim and Goldman, 2014*) the stabilization slows down the dynamics of all neurons in the network, thereby slowing down the relaxation of any deformation in the shape of the activity bump – not only the position of the activity bump on the ring attractor. In contrast, within the architecture considered here, the unimodal shape of the activity bump is maintained, while the velocity feedback mechanism slows down only noise driven diffusion of its position. Thus, the velocity coupling mechanism identified in this work may be relevant to the stabilization of short-term memory in head directions cells of rodents (*Taube, 2007*) and insects (*Seelig and Jayaraman, 2015*), where there is no evidence for slowly decaying deformations in the shape of the activity bump.

## Experimental predictions

The grid spacing of a single module is determined by the coefficient that tunes how strongly activity is modulated by velocity ($\gamma_\mu$ in *Equation 21*): larger values of $\gamma_\mu$ lead to smaller grid spacing. Thus, in an uncoupled network the grid spacing ratios are determined by the coefficients $\gamma_\mu$. However, in the network of coupled modules the spacing ratios are determined primarily by the inter-module coupling parameters. Each one of the coefficients $\gamma_\mu$ influences all the grid spacings, but has little effect on the spacing ratios. For example, even if the $\gamma_\mu$s are identical in all modules, or if only one module receives a velocity input, all modules shift their states with a velocity ratio that matches the desired grid spacing ratio. An interesting prediction arises under a scenario in which one of the modules is disconnected from the others. This removes positive couplings from the other modules, but leaves the negative self coupling within the module intact. Hence, the disconnected module is expected to weaken its response to velocity inputs, and increase its grid spacing. Similarly, other grid spacings, of modules that were originally connected to the disconnected module, are expected to increase as well (see *Figure 4—figure supplement 1*).

The joint activity of grid cells can be expected to lie within a two-dimensional space when salient sensory cues are available to the animal, regardless of the existence of an inter-module coupling mechanism. The existence of a coupling mechanism must therefore be tested under conditions in which external sensory cues are weak (*Burak, 2014*). It is instructive to compare this goal with what has been learned about population activity within a single module (*Yoon et al., 2013*; *Fyhn et al., 2007*; *Allen et al., 2014*; *Trettel et al., 2019*; *Gardner et al., 2019*). In that context, simultaneous recordings from pairs of grid cells were highly informative, since grid cells from a single module exhibit strong correlations (or anti-correlations) in their joint spiking activity. The preservation of these correlations, under conditions in which the animal's sense of position is disrupted, supports an

interpretation that the correlations are maintained by recurrent connectivity within each module. In contrast, cells belonging to different modules are expected to fire together in some positions in space, and refrain from firing together in other parts of the environment. Averaged over motion in a large environment, cell pairs from different modules are expected to exhibit weak correlations in their activity, even if the updates of module phases are fully coordinated.

Analysis of spike correlations in grid cells from different modules (*Trettel et al., 2019*; *Gardner et al., 2019*) confirms this expectation. During free running, spike correlation functions of grid cells from different modules are much weaker than those observed within a module. Despite being weak, these correlations can be statistically significant. Their existence originates from the fact that in any specific environment, and especially in small enclosures, the firing fields of two grid cells with different spatial scales slightly favor correlated or uncorrelated firing, depending on the precise overlap between the spatial receptive fields of the two cells. During sleep, these weak correlations are significantly reduced (*Trettel et al., 2019*; *Gardner et al., 2019*). A possible interpretation of this result is that there is no coupling between modules during sleep. However, an alternative explanation is that the reduction in correlations reflects an increase in the repertoire of positions and environments represented in the sleep state: regions of joint firing and regions of disjoint firing are expected to average out more evenly under such circumstances, leading to weaker correlations.

In order to test for the existence of a velocity coupling mechanism, it is desirable to test for correlations in the updates of phases of different modules, instead of directly testing for correlations in their phases. This will require simultaneous recordings from multiple grid cells, of sufficient numbers that will enable reliable tracking of the module phases. Appropriate recordings are not yet available, but techniques that enable simultaneous monitoring of large neural populations of the MEC (*Jun et al., 2017*; *Gu et al., 2018*; *Obenhaus et al., 2018*) are likely to enable their acquisition in the coming years. In similarity to the experiments that provided insights on the low dimensionality of activity within each module, it will be necessary to test for inter-module coupling under conditions in which the animal's internal sense of position is not anchored to salient external cues.

Our model assumes that grid cells in the MEC are involved in idiothetic path integration, and harnesses ingredients from models of path integration in the grid cell system to generate the coupling between modules. It is widely hypothesized that grid cells are indeed involved in path integration (*Hafting et al., 2005*; *McNaughton et al., 2006*; *Moser et al., 2008*; *Burak, 2014*), but this involvement is not experimentally established (see, however, *Gil et al., 2018*). Accordingly, a specific role of any particular cell type within the MEC in idiothetic path integration is not yet identified. A specific population of cells that may provide the substrate for the connectivity proposed in our model are the conjunctive cells observed mostly in layer III and deeper layers of the MEC (*Sargolini et al., 2006*), which play a pivotal role in models of path integration in the grid cell system. We note that these cells are tuned to head direction more closely than heading (*Raudies et al., 2015*), a difficulty that faces all models of path integration within the MEC. The resolution of this difficulty may involve computational elements within the entorhinal circuitry that have not yet been identified. Thus, future experimental findings concerned with the mechanisms underlying path integration may call for (and enable) corresponding refinements of our model.

Very little is known about synaptic connectivity between grid cells in the MEC, especially for cells belonging functionally to different modules. An important conclusion of our work is that synaptic connectivity between different modules may be beneficial for dynamically stabilizing the grid cell representation during path integration and memory maintenance. The specific form of connectivity that we identify is appealing for several reasons: first, it involves broad, relatively unstructured connectivity between grid cells, that depends only on their preferred heading preference. A second appealing feature of our proposed architecture is that it is sufficient to couple grid cells from modules with adjacent spacings, to achieve the desired stabilization of the grid cell representation. Since there is a relationship between grid spacing and position along the dorsal-ventral axis (*Hafting et al., 2005*; *Stensola et al., 2012*), all-to-all couplings between modules would require long-range connectivity within the MEC. Recent evidence (*Fu et al., 2018*) hints that synaptic connections between excitatory cells in the MEC may be more limited in range, but of sufficient spatial extent to allow for coupling of adjacent modules.

## Materials and methods

### Model and simulations details

The dynamics of the network are described by *Equation 3* (or by *Equation 22* for the Poisson spiking neuron case). The synaptic time constant $\tau = 10\,\text{ms}$, the transfer function $\phi(x) = \tau^{-1}\max(x, 0)$, and $I_0 = 3$. All simulations were done using the Euler method for integration, with a time step $dt = 0.1\,\text{ms}$.

### One-dimensional module

In the one-dimensional simulations the synaptic activation vector (*Equation 2*) includes synapses of the right and left sub-populations, each comprising $N = 1000$ neurons. Each neuron has a preferred phase $\theta_i \in [0, 1]$, uniformly arranged on a ring. The connectivity matrix $W$ is defined by

$$W = \begin{pmatrix} W^+ & W^- \\ W^+ & W^- \end{pmatrix}, \tag{8}$$

where

$$W_{ij}^{\pm} = w\left(|\theta_i - \theta_j \mp \varphi|_{\text{P}}\right), \quad w(\theta) = \frac{A}{2N}\left[\exp\left(-\frac{\theta^2}{2\sigma^2}\right) - 1\right]. \tag{9}$$

$W^{\pm}$ is a $N \times N$ matrix, $\varphi = 0.2$, $A = 200$, $\sigma^2 = 0.1$, and $|x_1 - x_2|_{\text{P}}$ is the minimal distance between two points $\{x_1, x_2\} \in [0, 1]$ with periodic boundary conditions on $[0, 1]$, namely

$$|x|_{\text{P}} = \min\left\{|x|_{mod1}, 1 - |x|_{mod1}\right\}. \tag{10}$$

### Coupled modules

Consider $m$ coupled modules. The firing rate (*Equation 3*) of neuron $i$ from module $\mu$ is (*Figure 2E*):

$$r_{\mu,i} = \phi\left(\sum_{j=1}^{2N} W_{ij}s_{\mu,j} + I_0 \pm b_\mu \pm a\sum_{\rho=1}^{m} C_{\mu\rho}\omega_\rho\right), \tag{11}$$

where $\omega_\mu$ is the velocity estimation of module μ (*Equation 4*) , $C_{\mu\rho}$ is the coupling strength from module $\rho$ to module μ ($C_{\mu\mu}$ is the self coupling strength of module μ), and the sign $\pm$ is equal to $+$ $(-)$ if the neuron $i$ belongs to a right (left) sub-population. The proportionality factor $a = \left[\beta\sum_{i=1}^{2N}\phi'\left(\sum_{j=1}^{2N} W_{ij}\bar{s}_j + I_0\right)\right]^{-1}$ is included to simplify the units of the coupling strengths $C_{\mu\rho}$. Note that $a \cdot \omega_\mu$ does not depend on the parameter β. Thus, the choice of β is of no consequence for the dynamics, and this parameter is included only for the sake of derivation convenience.

The coupling between the modules can be interpreted as arising from synaptic connectivity. This can be seen by re-writing *Equation 11* as:

$$r_{\mu,i} = \phi\left(\sum_{j=1}^{2N} W_{ij}s_{\mu,j} + a\sum_{\rho=1}^{m} C_{\mu\rho}\sum_{j=1}^{2N} W_{ij}^c s_{\mu,j} + I_0 \pm b_\mu\right), \tag{12}$$

where the $2N \times 2N$ coupling connectivity matrix is:

$$W^c = \frac{\beta}{\tau}\begin{pmatrix} 1 & \cdots & 1 & -1 & \cdots & -1 \\ \vdots & & \vdots & \vdots & & \vdots \\ 1 & \cdots & 1 & -1 & \cdots & -1 \\ -1 & \cdots & -1 & 1 & \cdots & 1 \\ \vdots & & \vdots & \vdots & & \vdots \\ -1 & \cdots & -1 & 1 & \cdots & 1 \end{pmatrix} \tag{13}$$

Thus, the synapses responsible for the coupling of any two neurons belonging to modules $\mu$ and $\rho$ are of the same magnitude, $a\beta C_{\mu\rho}/\tau$ , and their signs depend only on the sub-populations (left or right) of the pre- and post-synaptic neurons. In the case of $m = 2$, we use the coupling parameters:

$Cs = -20$, $C_{12} = C_1 \approx 14.14$ and $C_{21} = C_2 \approx 28.3$. In the case of $m = 3$, we use the parameters: $Cs = -20$, $C_{12} \approx 14.14$, $C_{21} = C_{23} \approx 9.4$, $C_{32} \approx 28.3$, and $C_{13} = C_{31} = 0$. Additional details on the choice of coupling parameters are provided in Appendices 2 and 3.

## Two-dimensional modules

In two dimensions, each module contains four sub-populations, and the synaptic activation vector is:

$$\vec{s} = \begin{pmatrix} \vec{s}_R \\ \vec{s}_L \\ \vec{s}_U \\ \vec{s}_D \end{pmatrix} \tag{14}$$

Each sub-population contains $N^2 = 64^2$ neurons, arranged on a parallelogram. The preferred phase of the $i$'th neuron in each sub-population is:

$$\vec{\theta}_i = \begin{pmatrix} \theta_{x_i} \\ \theta_{y_i} \end{pmatrix} = x_i \vec{u_1} + y_i \vec{u_2}, \quad \vec{u}_1 = \begin{pmatrix} 1 \\ 0 \end{pmatrix}, \quad \vec{u}_2 = \begin{pmatrix} 0.5 \\ \frac{\sqrt{3}}{2} \end{pmatrix}. \tag{15}$$

where $x_i$ and $y_i$ are distributed uniformly in the interval $[0, 1]$. The connectivity matrix is now:

$$W = \begin{pmatrix} W^R & W^L & W^U & W^D \\ W^R & W^L & W^U & W^D \\ W^R & W^L & W^U & W^D \\ W^R & W^L & W^U & W^D \end{pmatrix}, \tag{16}$$

where

$$W_{ij}^{R,L} = w\left( \left| \vec{\theta}_i - \vec{\theta}_j \mp \begin{pmatrix} \varphi \\ 0 \end{pmatrix} \right|_{P_2} \right), \quad W_{ij}^{U,D} = w\left( \left| \vec{\theta}_i - \vec{\theta}_j \mp \begin{pmatrix} 0 \\ \varphi \end{pmatrix} \right|_{P_2} \right), \tag{17}$$

and

$$w(\theta) = \frac{A}{4N^2} \left[ \exp\left( -\frac{\theta^2}{2\sigma^2} \right) - 1 \right]. \tag{18}$$

The distance measure $|\cdot|_{P_2}$ is defined using periodic boundary conditions on the parallelogram (see *Figure 4A*): $|\vec{x}_1 - \vec{x}_2|_{P_2}$ is the minimal distance between the two points $\vec{x}_1$ and $\vec{x}_2$ on the torus that is created by gluing the opposite edges of the parallelogram defined by the vertices $(0, 0), (1, 0), (\frac{1}{2}, \frac{\sqrt{3}}{2}), (\frac{3}{2}, \frac{\sqrt{3}}{2})$ (*Figure 4A*, compare with *Equation 10*, used in the one-dimensional case).

The firing rate of neuron $i \in$ {sub-population $R$ or $L$} from module $\mu$ is:

$$r_{\mu,i} = \phi\left( \sum_{j=1}^{4N^2} W_{ij} s_{\mu,j} + I_0 \pm b_{\mu,x}(t) \pm a \sum_{\rho=1}^{m} C_{\mu\rho} \omega_{\rho,x} \right), \tag{19}$$

where

$$\omega_{\mu,x} \equiv \frac{\beta}{\tau} \left( \sum_{i \in R} s_{\mu,i} - \sum_{i \in L} s_{\mu,i} \right). \tag{20}$$

Firing rates of neurons from the up and down sub-populations are obtained from *Equation 19*-Equation 20 by replacing $x \rightarrow y$, , $R \rightarrow U$, and $L \rightarrow D$.

In each module, responses to horizontal and vertical velocity inputs are independent: the right and left sub-populations respond to the horizontal velocity inputs, and affect $\theta_x$, while the up and down sub-populations respond to the vertical velocity inputs and affect $\theta_y$. Hence, the two-dimensional response tensor separates into independent, horizontal and vertical components with the same structure as in the one-dimensional case. Since the linear response tensor (in each direction) is

identical to that of the one-dimensional case, the coupling parameters are chosen in the same way in one and two dimensions.

## External velocity input

The external velocity input to module $\mu$ (in two dimensions) in $q \in \{x, y\}$ direction is (*Equation 19*):

$$b_{\mu,q}(t) = \gamma_\mu \left( V_q(t) + \eta_{\mu,q}(t) \right). \tag{21}$$

$\gamma_\mu$ is a proportionality factor that depends on the module, $V_q(t)$ is a the component of the animal's velocity in the $q$ direction, and $\eta_{\mu,q}(t)$ is a white noise process with $\langle \eta_{\mu,q}(t) \eta_{\rho,q'}(t') \rangle = \eta^2 \delta_{qq'} \delta_{\mu\rho} \delta(t - t')$. In *Figure 4B* and *Figure 6B–C* the external input is not noisy, so $\eta = 0$. In *Figure 4D–E* and *Figure 5*, $\eta = 0.02 \,\mathrm{m} \cdot \mathrm{s}^{-0.5}$.

In *Figure 3E–F* (one dimension) $\gamma_1 = 0.06$ and $\gamma_2 = 0$. In the simulations of *Figure 4*; *Figure 6* (two dimensions) $\gamma_1 = 0.06$ , $\gamma_2 = \lambda 0.06$ , $\gamma_3 = \lambda^2 0.06$. Thus, even without coupling of the modules, the spacing ratio $\lambda$ is achieved by the ratios of the inputs strengths $\gamma_\mu$, and therefore, we can compare between the coupled and uncoupled phases and readout (*Figure 4*; *Figure 6*).

## Spiking network

In the case of spiking Poisson neurons *Equation 3* is replaced by:

$$\dot{s}_i + \frac{s_i}{\tau} = \sum_\chi \delta(t - t_i^\chi), \tag{22}$$

where

$$\sum_\chi \delta(t - t_i^\chi) \tag{23}$$

is the spike train of neuron $i$, and $t_i^\chi$ are the spike times. Each neuron $i$ generates spikes sampled from a Poisson distribution with a firing rate $r_i(t)$, as defined in *Equation 3* (*Stevens and Zador, 1996*; *Gerstner and Kistler, 2002*; *Burak and Fiete, 2012*).

## Decoding

Decoding of the animal's trajectory, based on spike trains, is performed in *Figure 5*; *Figure 6* using a decoder that sums spikes from recent history with an exponential temporal kernel (*Mosheiff et al., 2017*). In *Figure 5*, we simulate a spike train for each neuron (*Equation 23*), sampled from an inhomogeneous Poisson process with a firing rate $r_i(t)$ (note that the the network dynamics are deterministic and spikes are used only in the readout process). In *Figure 6*, the stochastic spike train is part of of the dynamics (*Equation 22*). In both cases, the decoded location of the animal in time $t$ is (Eqs. S10, S11 and S13 in *Mosheiff et al., 2017*):

$$\hat{\vec{x}}(t) = \mathrm{argmax}_{\vec{x}} \sum_i \ln[\bar{r}_i(\vec{x})] \int_{-\infty}^t dt' \exp\left( -\frac{t - t'}{\tau_d} \right) \sum_\chi \delta(t' - t_i^\chi). \tag{24}$$

The summation is over all neurons. Here, $\tau_d = 10 \,\mathrm{ms}$ for all modules. The integral in *Equation 24* yields an effective spike count of neuron $i$, weighted in time using a decaying exponential kernel, and $\bar{r}_i(\vec{x})$ is the receptive field of neuron $i$ at location $\vec{x}$, measured separately from the firing rate of each neuron in the steady state of the dynamics.

## Diffusion tensor

Consider a system of spiking Poisson neurons, and $m$ one-dimensional modules. The internal noise introduces a diffusive drift. The network is now a single continuous attractor with dimension $m$ (see Appendix 5). Hence, the diffusion tensor is a $m \times m$ matrix, that can be calculated using Eq. S24 in *Burak and Fiete (2012)*, for the dynamics of the $m$ dimensional attractor (*Equation 77*, *Equation 78* and *Equation 79* in Appendix 5):

$$D_{\mu\rho}(\vec{\theta}) = \frac{1}{2}\sum_{i=1}^{2Nm} \nu_{\mu,i}(\vec{\theta})\nu_{\rho,i}(\vec{\theta})\bar{r}_i(\vec{\theta}). \tag{25}$$

where the summation is over all neurons, $\bar{r}_i(\vec{\theta})$ is the firing rate of neuron $i$ in the steady state of the system, and $\nu_\mu(\vec{\theta})$ is the left null eigenvector of the dynamics (*Equation 78*) corresponding to a phase shift in the direction of module $\mu$ (calculated numerically).

## Acknowledgements

This research was supported by the Israel Science Foundation grant No. 1745/18 and (in part) by grant No. 1978/13. We acknowledge support from the Gatsby Charitable Foundation, and from the Dalia and Dan Maydan Fellowship (NM).

## Additional information

### Funding

| Funder | Grant reference number | Author |
|---|---|---|
| Israel Science Foundation | 1745/18 | Yoram Burak |
| Israel Science Foundation | 1978/13 | Yoram Burak |
| Gatsby Charitable Foundation | | Yoram Burak |
| Dalia and Dan Maydan Fellowship | | Noga Mosheiff |

The funders had no role in study design, data collection and interpretation, or the decision to submit the work for publication.

### Author contributions

Noga Mosheiff, Conceptualization, Formal analysis, Funding acquisition, Investigation, Methodology, Writing—original draft, Project administration, Writing—review and editing; Yoram Burak, Conceptualization, Resources, Formal analysis, Supervision, Funding acquisition, Investigation, Methodology, Writing—original draft, Project administration, Writing—review and editing

### Author ORCIDs

Noga Mosheiff (iD) http://orcid.org/0000-0002-3649-4183
Yoram Burak (iD) https://orcid.org/0000-0003-1198-8782

### Decision letter and Author response

Decision letter https://doi.org/10.7554/eLife.48494.019
Author response https://doi.org/10.7554/eLife.48494.020

## Additional files

### Supplementary files

• Transparent reporting form
DOI: https://doi.org/10.7554/eLife.48494.011

### Data availability

This is a theoretical work. There are no data sets associated with it.

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

## Appendix 1

DOI: https://doi.org/10.7554/eLife.48494.012

### Readout of the phase velocity

In this Appendix, we derive the relationship between $\omega$ (**Equation 4**) and the phase velocity $\dot{\theta}$ (**Equation 1**) . To simplify the presentation, we consider separately two situations: first, a noise free network, in which the phase velocity is driven by velocity inputs. Second, a module consisting of Poisson spiking neurons where, for simplicity, we set the velocity inputs to be zero. In the latter case the phase velocity is driven by the stochasticity of the neurons participating in the network dynamics. From the derivation it is easily seen that in the general case, and as long as the linearization of the dynamics (forming the basis of the calculation in both cases) is valid, the phase velocity, as well as $\omega$ can be expressed as a sum of independent contributions arising from the two sources considered below.

### Phase velocity due to velocity inputs

Consider a single one dimensional module, whose dynamics follow **Equation 3**, and whose synaptic connectivity is described by the double ring model, as specified by **Equations 8-10**. We expand the dynamic equation around a steady state $\bar{s}(\theta)$, so that $s_i = \bar{s}_i(\theta) + \delta s_i$, and assume small velocity input $dI(t)$ (see similar expansion in **Burak and Fiete, 2012**). After linearization:

$$\dot{\delta s_i} = \sum_{j=1}^{2N} K_{ij}(\theta)\delta s_j \pm dI(t)\phi'(\bar{g}_i(\theta)) \tag{26}$$

where

$$K_{ij}(\theta) = -\frac{1}{\tau}\delta_{ij} + \phi'(\bar{g}_i(\theta))W_{ij} \tag{27}$$

and $\bar{g}_i(\theta) = \sum_{j=1}^{2N} W_{ij}\bar{s}_j + I_0$ is the synaptic input in the steady state $\bar{s}_i(\theta)$. The velocity of the phase $\theta$ is obtained by projection of **Equation 26** on the left eigenvector of $K$ with zero eigenvalue, $\vec{v}(\theta)$ (**Burak and Fiete, 2012**). After some algebra we obtain

$$\dot{\theta} = v^T \cdot \dot{\delta s} = \alpha dI(t), \tag{28}$$

where

$$\alpha = \left(\sum_{i\in R} v_i\phi'(\bar{g}_i) - \sum_{i\in L} v_i\phi'(\bar{g}_i)\right). \tag{29}$$

Hence, $\dot{\theta}$ is proportional to the velocity input $dI(t)$. Note that both $\vec{v}$ and $\bar{g}$ rotate together with the position $\theta$ of the activity bump. Thus, $\alpha$ is independent of $\theta$.

Let us define the $2N$ dimensional vector:

$$\vec{v}_0 = \beta \begin{pmatrix} 1 \\ \dots \\ 1 \\ -1 \\ \dots \\ -1 \end{pmatrix}, \tag{30}$$

where $\beta$ is a constant, whose value is determined below. The projection of **Equation 26** on $\vec{v}_0$ results in

$$v_0^T \cdot \delta\dot{\vec{s}} = -\frac{v_0^T \cdot \delta\vec{s}}{\tau} + \sum_{i,j=1}^{2N} v_{0,i}\phi'(\bar{g}_i)W_{ij}\delta s_j + dI(t)\sum_{i=1}^{2N}\frac{v_{0,i}^2}{\beta}\phi'(\bar{g}_i). \tag{31}$$

The second term on the right hand side of **Equation 31** vanishes due to the symmetry of $\phi'(\bar{g}_i)$ to exchange of the right and left sub-populations, the antisymmetry of $v_0$ to this exchange, and the structure of the matrix $W$. Using the definition of $\omega$ (**Equation 4**), we can write:

$$\omega = \frac{v_0^T \cdot \vec{s}}{\tau} = \frac{v_0^T \cdot \delta\vec{s}}{\tau}. \tag{32}$$

Thus, **Equation 31** becomes:

$$\tau\dot{\omega} = -\omega + \tilde{\alpha}dI(t), \tag{33}$$

where

$$\tilde{\alpha} = 2\beta\sum_{i\in R}\phi'(\bar{g}_i). \tag{34}$$

In **Equation 34** we use again the symmetry of $\phi'(\bar{g}_i)$ to exchange of the right and left sub-populations. We set $\beta$ (**Equation 30**) such that $\tilde{\alpha} = \alpha$. Using **Equation 29** and **Equation 34**:

$$\beta = \frac{\sum_{i\in R}v_i\phi'(\bar{g}_i) - \sum_{i\in L}v_i\phi'(\bar{g}_i)}{2\sum_{i\in R}\phi'(\bar{g}_i)}. \tag{35}$$

From **Equation 33** we see that the readout velocity $\omega$ is a convolution of $\alpha dI(t)$, with the filter

$$f(t) = \frac{1}{\tau}\exp\left(-\frac{t}{\tau}\right). \tag{36}$$

Combining **Equation 28** and **Equation 33** we conclude that

$$\omega(t) = f(t) * \dot{\theta}(t). \tag{37}$$

Hence, $\omega$ is equal to the phase velocity, smoothed over a relatively short time scale set by the synaptic time constant ($\tau = 10\,\text{ms}$).

## Phase velocity due to internal noise

Consider next a single one dimensional module with Poisson spiking neurons, following the dynamics of **Equation 22**. In the limit of large firing rates, **Equation 22** can be replaced by

$$\dot{s}_i + \frac{s_i}{\tau} = r_i + \xi_i, \tag{38}$$

where $\xi_i$ is a white noise process with $\langle\xi_i(t)\xi_j(t')\rangle = r_i(t)\delta_{ij}\delta(t-t')$ (**Burak and Fiete, 2012**). We use a similar expansion as in section *Phase velocity due to velocity inputs* in this Appendix , and project the linearized dynamics on $\vec{v}(\theta)$, the left null eigenvector of $K$, to obtain:

$$\dot{\theta} = v^T \cdot \delta\dot{\vec{s}} = dI(t)\sum_{i=1}^{2N}\frac{v_iv_{0,i}}{\beta}\phi'(\bar{g}_i) + v^T \cdot \vec{\xi}. \tag{39}$$

The first term on the right hand side of **Equation 39** is the contribution to the phase velocity arising from the velocity inputs, discussed above. From here on we assume that the velocity input vanishes. The only contribution to the phase velocity is then the second term of **Equation 39**, originating from the neural stochasticity. In order to relate this term to $\omega$, let us define:

$$\vec{v}(\theta) = \vec{v}_+ + \vec{v}_- \tag{40}$$

$$\vec{v}_+ = \frac{1}{2}\begin{pmatrix} \vec{v}_L + \vec{v}_R \\ \vec{v}_L + \vec{v}_R \end{pmatrix}, \; \vec{v}_- = \frac{1}{2}\begin{pmatrix} \vec{v}_R + \vec{v}_L \\ \vec{v}_L + \vec{v}_R \end{pmatrix} \tag{41}$$

where $\vec{v}_R$ and $\vec{v}_L$ are the first or last $N$ coordinates of $\vec{v}(\theta)$, respectively (**Appendix 1—figure 1**). **Equation 39** becomes:

$$\dot{\theta} = (v_-^T + v_+^T) \cdot \vec{\xi}. \tag{42}$$

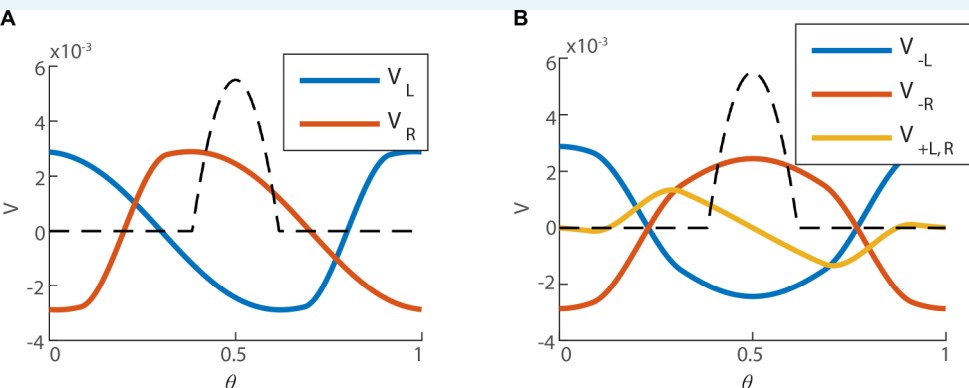

**Appendix 1—figure 1.** Structure of the left null eigenvector. (**A**) The left eigenvector of $K$ with zero eigenvalue, $\vec{v}(\theta = 0.5)$ (the bump is centered around $\theta = 0.5$), computed numerically. The first $N$ coordinates of $\vec{v}$ (right sub-population) in red, and the last $N$ coordinates of $\vec{v}$ (left sub-population) in blue. (**B**) $\vec{v}_-$ (red and blue for the right and left sub-population respectively), and $\vec{v}_+$ (yellow, identical for both of the sub-populations), as defined in **Equation 41**. The black dashed line represents the firing rate bump of the steady state (unitless for the sake of comparison with $\vec{v}$).

DOI: https://doi.org/10.7554/eLife.48494.013

Next, we examine the structure of $\vec{v}_+$ and $\vec{v}_-$. The left and right components of $v_+$ are spatially antisymmetric with respect to the peak of the activity bump, and precisely vanish at the center of the bump (yellow trace in **Appendix 1—figure 1B**). The components of $v_-$ are symmetric (blue and red traces). We note that the Poisson noise is expressed only within the relatively narrow extent of the activity bump (dashed line). Two consequences follow: first, $\vec{v}_+^T \cdot \vec{\xi}$ is very small due to the nearly vanishing values of $v_+$ on the activity bump. Second, the components of $v_-$ are nearly constant in magnitude along the narrow extent of the bump, but have opposite signs in the left and right sub-populations. This magnitude can be evaluated by rewriting **Equation 35** as:

$$\beta \sum_{i \in R} \phi'(\bar{g}_i) = \sum_{i \in R} v_{-i} \phi'(\bar{g}_i). \tag{43}$$

As the magnitude of $v_-$ is nearly constant along the bump, we conclude from **Equation 43** that this magnitude equals approximately $\beta$, and $\vec{v}_-^T \cdot \vec{\xi} \approx \vec{v}_0^T \cdot \vec{\xi}$. Hence, **Equation 42** becomes:

$$\dot{\theta} \approx v_0^T \cdot \vec{\xi}. \tag{44}$$

We now project the linearized dynamics of **Equation 38** on the vector $\vec{v}_0$ to obtain:

$$\tau\dot{\omega} = -\omega + v_0^T \cdot \vec{\xi} \approx -\omega + \dot{\theta}. \tag{45}$$

Thus, $\omega$ is approximately a convolution of the velocity phase with the exponential filer $f$, with the synaptic time integration $\tau$. Therefore:

$$\omega \approx f * \dot{\theta}, \tag{46}$$

as in the case of phase velocity due to external inputs.

# Appendix 2

DOI: https://doi.org/10.7554/eLife.48494.012

## Linear response tensor

We next wish to understand how our system of coupled modules responds to an external velocity input $\vec{b}(t)$. First, consider two modules, each in one dimension, that follow the dynamics of **Equation 3**. The synaptic activation vector is now $4N$ dimensional and has the form:

$$\vec{s} = \begin{pmatrix} \vec{s}_1 \\ \vec{s}_2 \end{pmatrix} = \begin{pmatrix} \vec{s}_{1,R} \\ \vec{s}_{1,L} \\ \vec{s}_{2,R} \\ \vec{s}_{2,L} \end{pmatrix}. \tag{47}$$

The firing rate of neuron $i$ from module 1 is (**Figure 2E**):

$$r_{1,i} = \phi \left( \sum_{j=1}^{2N} W_{ij} s_{1,j} + I_0 \pm b_1 \pm a C_1 \omega_2 \pm a C_s \omega_1 \right), \tag{48}$$

where the total velocity input is $dI = b_1 + a C_1 \omega_2 + a C_s \omega_1$. A similar equation for the rate $r_{2,i}$ is obtained by switching the indices $1 \leftrightarrow 2$ in **Equation 48**. The constant

$$a = \frac{1}{\beta \sum_{i=1}^{2N} \phi'(\bar{g}_i)} = \frac{1}{\alpha} \tag{49}$$

is a proportionality factor that could in principle be absorbed into the definition of the coupling strength parameters, but makes the units of the coupling strengths $C_\mu$ more convenient. Substitution of $dI$ and **Equation 37** in **Equation 28** yields:

$$\dot{\theta}_1 = \alpha b_1 + C_1 f * \dot{\theta}_2 + C_s f * \dot{\theta}_1 . \tag{50}$$

More generally, for $m$ modules:

$$\dot{\vec{\theta}} = \alpha \vec{b}(t) + f * C \dot{\vec{\theta}}. \tag{51}$$

Thus, the phase velocities are proportional to the external velocity input, plus the velocities of all other modules filtered in time, and coupled by the matrix C. If we assume that the velocity is sufficiently small, that the position can be regarded as fixed within the synaptic time scale, the convolution with the filter $f$ can be omitted (note that the integral of $f$, **Equation 36**, is equal to unity), and we obtain:

$$\dot{\vec{\theta}} = \alpha (I - C)^{-1} \cdot \vec{b}(t) \equiv X \cdot \vec{b}(t), \tag{52}$$

where $X$ is the linear response tensor.

## B.1 Two modules

In the case of two modules and identical self coupling strength $C_s$ for both modules, we can easily find the eigenvalues $X_\pm$ and eigenvectors of $X$. The coupling matrix in this case is:

$$C = \begin{pmatrix} C_s & C_1 \\ C_2 & C_s \end{pmatrix}. \tag{53}$$

Using **Equation 7** we obtain:

$$X_\pm = \frac{\alpha}{1 - C_\pm} = \frac{\alpha}{1 - (C_s \pm \sqrt{C_1 C_2})} \, . \tag{54}$$

The eigenvectors of $X$ are:

$$\vec{u}_\pm = \begin{pmatrix} \sqrt{C_1} \\ \pm\sqrt{C_2} \end{pmatrix} . \tag{55}$$

Thus, the phase motion of the two modules can be represented as:

$$\begin{pmatrix} \dot{\theta}_+ \\ \dot{\theta}_- \end{pmatrix} = \begin{pmatrix} X_+ & 0 \\ 0 & X_- \end{pmatrix} \cdot \begin{pmatrix} b_+ \\ b_- \end{pmatrix}, \tag{56}$$

where

$$\begin{pmatrix} \theta_+ \\ \theta_- \end{pmatrix} = \frac{1}{2\sqrt{C_1 C_2}} \begin{pmatrix} \sqrt{C_2}\theta_1 + \sqrt{C_1}\theta_2 \\ \sqrt{C_2}\theta_1 - \sqrt{C_1}\theta_2 \end{pmatrix}, \tag{57}$$

and

$$\begin{pmatrix} b_+ \\ b_- \end{pmatrix} = \frac{1}{2\sqrt{C_1 C_2}} \begin{pmatrix} \sqrt{C_2}b_1 + \sqrt{C_1}b_2 \\ \sqrt{C_2}b_1 - \sqrt{C_1}b_2 \end{pmatrix} . \tag{58}$$

Here, $\theta_\pm$ are the joint or relative phases of the modules, respectively. We choose parameters such that $X_+$ is large and $X_-$ is small in order to obtain large joint motion and small relative motion between modules for any external input. Note that $X_- \approx 0$ implies that $\dot{\theta}_- \approx 0$ (*Equation 56*), and in this case (*Equation 57*):

$$\frac{\dot{\theta}_1}{\dot{\theta}_2} \approx \sqrt{\frac{C_1}{C_2}} . \tag{59}$$

Thus, we can set the velocity ratio between modules (namely, the spacing ratio $\lambda$), by the ratio of the coupling parameters, using *Equation 59*. We choose

$$\sqrt{\frac{C_2}{C_1}} = \lambda = \sqrt{2} \, . \tag{60}$$

We choose $X_+ = \alpha$ to maintain the same response to coordinated velocity inputs, as in the case of no coupling. Using *Equation 54* we see that this requires $C_s + \sqrt{C_1 C_2} = 0$, which implies

$$\begin{aligned} C_1 &= -C_s/\lambda \\ C_1 &= -\lambda C_s, \end{aligned} \tag{61}$$

and

$$X_- = \frac{\alpha}{1 - 2C_s} \tag{62}$$

To obtain small $X_-$, $C_s$ should be negative, with a large absolute magnitude. Note that a large positive magnitude of $C_s$, although seems here as a suitable choice, would cause instability of the system (discussed in Appendix D).

# Appendix 3

DOI: https://doi.org/10.7554/eLife.48494.012

## Optimization of the coupling strengths for *m* modules

We now wish to find appropriate coupling strengths, $C_{\mu\rho}$, in the general case of $m$ modules, while allowing for different self couplings in each module. We optimize $C_{\mu\rho}$ as follows.

Let $\vec{u}$ be a vector in the direction of the mutual motion of $\vec{\theta}$. The ratios of the components of $\vec{u}$ are the ratios between corresponding grid spacings. We demand that $\vec{u}$ is an eigenvector of $X$ (**Equation 7**) with an eigenvalue $\alpha$. Under this choice, the motion response in the mutual direction is the same as without coupling (where $\dot{\vec{\theta}} = \alpha\vec{b}$, **Equation 52**). Any other eigenvalue of $X$ should be close to zero, such that the motion in any direction, other than the mutual direction, is small.

Equivalent requirements are that $\vec{u}$ is an eigenvector of $C$ with eigenvalue , and any other eigenvalue of $C$ has a large magnitude (see **Equation 52**). In addition, as discussed below (*Stability conditions*) all the eigenvalues of $C$ must be smaller than unity, which excludes the possibility that they take large positive values. In order to find the optimal coupling parameters we apply the Karush-Kuhn-Tucker theorem with the Lagrangian:

$$\mathcal{L} = Tr(C) + \sum_{\mu=1}^{m} \epsilon_i \sum_{\rho=1}^{m} C_{\mu\rho} u_\rho + \sum_{\mu=1}^{m} \zeta_\mu (C_{\mu\mu}^2 - Q). \tag{63}$$

The first term of the Lagrangian favors negative eigenvalues with large absolute magnitude, whereas the second term, involving $m$ Lagrange multipliers $\epsilon_i$, enforces the existence of the eigenvector $\vec{u}$ with eigenvalue . As the coupling magnitudes cannot be infinitely large, we constrain the self couplings by requiring that $C_{\mu\mu}^2 < Q$ , where $Q > 0$ is a constant. This constraint is enforced by the third term in **Equation 63**, where $\zeta_\mu \geq 0$'s are Karush-Kuhn-Tucker multipliers. We thus obtain the equations:

$$\frac{\partial \mathcal{L}}{\partial C_{\mu\rho}} = 0, \tag{64}$$

$$\frac{\partial \mathcal{L}}{\partial \epsilon_i} = 0, \tag{65}$$

$$\zeta_\mu \left( C_{\mu\mu}^2 - Q \right) = 0, \tag{66}$$

which results in:

$$C_{\mu\mu} = -\sqrt{Q} \equiv C_s, \quad C\vec{u} = 0. \tag{67}$$

For $m = 2$ and $\vec{u} = \begin{pmatrix} 1 \\ \lambda \end{pmatrix}$ we obtain a single solution:

$$C = \begin{pmatrix} C_s & -C_s/\lambda \\ -\lambda C_s & C_s \end{pmatrix}, \tag{68}$$

which is identical to the solution we found in Appendix B.1.

For $m > 2$ and a given vector $\vec{u}$ there are multiple solutions. The general solution follows the equations:

$$C_{\mu\mu} = C_s \tag{69}$$

$$\sum_{\rho} C_{\mu\rho} u_{\rho} = 0, \tag{70}$$

for every $\mu$. As there are $m^2$ parameters and $2m$ equations to follow, there are $m^2 - 2m$ degrees of freedom in the solution.

For example, for $m = 3$ and $u_i = \lambda^{i-1}$, the solutions are the all $C_{\mu\rho}$'s that obey:

$$C_s + \lambda C_{12} + \lambda^2 C_{13} = 0, \tag{71}$$

$$C_{21} + \lambda C_s + \lambda^2 C_{23} = 0, \tag{72}$$

$$C_{31} + \lambda C_{32} + \lambda^2 C_s = 0. \tag{73}$$

Making the additional choice that only successive modules are coupled yields the solution:

$$C = \begin{pmatrix} C_s & -C_s/\lambda & 0 \\ -\lambda C_s - \lambda^2 C_{23} & C_s & C_{23} \\ 0 & -\lambda C_s & C_s \end{pmatrix}, \tag{74}$$

where there is a freedom in choosing $C_{23}$. In **Figure 4**, **Figure 5** and **Figure 6** we chose $C_{23} = C_{21}$.

For general number of modules $m$, coupling only successive modules leads to a degeneracy of solutions with $m - 2$ degrees of freedom, since the matrix $C_{\mu\rho}$ includes $3m - 2$ non-vanishing entries and we solve $2m$ equations. The remaining degeneracy can be resolved by requiring identical coupling strengths to each module ($1 < \mu < m$) from its neighbors: $C_{\mu,\mu+1} = C_{\mu,\mu-1}$. The solution is then:

$$C = \begin{pmatrix} C_s & -C_s/\lambda & 0 & \cdots & & 0 \\ C_0 & C_s & C_0 & \ddots & & \vdots \\ 0 & C_0 & C_s & C_0 & \ddots & \vdots \\ \vdots & \ddots & C_0 & C_s & C_0 & 0 \\ \vdots & & \ddots & C_0 & C_s & C_0 \\ 0 & \cdots & & 0 & -\lambda C_s & C_s \end{pmatrix}, \tag{75}$$

where $C_0 = -\lambda C_s/(1 + \lambda^2)$.

In principle, other constraints on the coupling strengths can be used instead of the requirement that $C_{\mu\mu}^2 < Q$ for all $\mu$. For example, limiting the magnitude of all the coupling strengths ($C_{\mu\rho}^2 < Q$), instead of only the magnitude of the self couplings, yields solutions with smaller range of allowed parameters (in the case of multiple solutions), but otherwise identical structure. A different possible constraint is of the form $\sum_{\mu,\rho=1}^{m} C_{\mu\rho}^2 < Q$. This constraint results in a symmetric solution: $C_{\mu\rho} \propto \frac{u_{\mu} u_{\rho}}{\sum_k u_k^2} - \delta_{\mu\rho}$. We prefer the constraint in **Equation 63** as we do not see a compelling reason to limit all of the coupling parameters together instead of limiting them separately.

## Appendix 4

DOI: https://doi.org/10.7554/eLife.48494.012

### Stability conditions

To map the stability conditions of the coupled network, we consider the dynamics of the velocity estimation $\vec{\omega}$ (**Equation 33**). As **Equation 33** is obtained by projecting the full dynamics of the network on the vector $\vec{v}_0$, stable network would result in finite $\vec{\omega}$. We substitute the total velocity input $\vec{dI} = aC \cdot \vec{\omega} + \vec{b}(t)$ in **Equation 33** and obtain:

$$\tau \dot{\vec{\omega}} = -\vec{\omega} + C \cdot \vec{\omega} + \alpha \vec{b}(t) = -(I - C) \cdot \vec{\omega} + \alpha \vec{b}(t). \tag{76}$$

Thus, in order to maintain the stability of $\vec{\omega}$, as well as the stability of the full network, all eigenvalues of $C$ must be smaller than unity, and all eigenvalues of $X$ (**Equation 7**) must be positive.

# Appendix 5

DOI: https://doi.org/10.7554/eLife.48494.012

## The coupled network as a single attractor

In this Appendix we consider the system as a single neural network with recurrent connectivity that includes both the inter- and intra-modular synaptic connections, instead of thinking of the system as composed of $m$ coupled attractors (modules). The derivation of the system dynamics presented here, is necessary in order to calculate the diffusion tensor (**Equation 25**). In addition, it offers an alternative approach for obtaining the linear response tensor $X$, and provides some additional insights on the behaviour of the network.

We expand the dynamics of **Equation 3** around the steady state $\bar{s}(\vec{\theta})$ (see similar expansion in Appendix A). Here $\vec{\theta}$ is the vector of represented phases in all modules, where $\theta_\mu$ is the phase of module $\mu$. The dynamics of neuron $i$ from module $\mu$ (for simplicity, we assume one dimensional modules) can be written as:

$$\delta \dot{s}_i = \sum_{j=1}^{2Nm} K_{m,ij} \cdot \delta s_j \pm b_\mu \phi' \left[ \bar{g}_i(\theta_\mu) \right],\tag{77}$$

where the sign $\pm$ is $+$ () if neuron $i$ belongs to a right (left) sub-population. Here, $K_m$ is a $2Nm \times 2Nm$ matrix:

$$K_m = \begin{pmatrix} K(\theta_1) + B_{11} & B_{12} & \cdots & B_{1m} \\ B_{21} & K(\theta_2) + B_{22} & & \vdots \\ \vdots & & K(\theta_{m-1}) + B_{m-1,m-1} & B_{m-1,m} \\ B_{m1} & \cdots & B_{m,m-1} & K(\theta_m) + B_{mm} \end{pmatrix},\tag{78}$$

where $K(\theta)$ is defined in **Equation 27**, and

$$B_{\mu\rho} = \frac{a C_{\mu\rho} \beta}{\tau} \begin{pmatrix} -\Phi'_\mu & \Phi'_\mu \\ \Phi'_\mu & -\Phi'_\mu \end{pmatrix}, \quad \Phi'_\mu = \begin{pmatrix} \phi'(\bar{g}_1(\theta_\mu)) & \cdots & \phi'(\bar{g}_1(\theta_\mu)) \\ \vdots & \ddots & \vdots \\ \phi'(\bar{g}_N(\theta_\mu)) & \cdots & \phi'(\bar{g}_N(\theta_\mu)) \end{pmatrix}.\tag{79}$$

As the system is a continuous attractor of dimension $m$, there exist $m$ left eigenvectors of $K_m$ with zero eigenvalue, $\vec{v}_\mu(\vec{\theta})$, which we compute numerically. The diffusion tensor is computed in **Equation 25**, using these vectors $\vec{v}_\mu(\vec{\theta})$.

By projecting **Equation 77** on $\vec{v}_\mu(\vec{\theta})$ we obtain the linear response tensor $X$, such that **Equation 6** is valid. Explicitly:

$$X_{\mu\rho} = \sum_{i\in\rho} \nu_{\mu,i} \delta I_i \phi'(\bar{g}_i(\theta_\mu)),\tag{80}$$

where,

$$\delta I_i = \begin{cases} -1 & i \in \{\text{left}\} \\ 1 & i \in \{\text{right}\} \end{cases},\tag{81}$$

and $i \in \rho$ means that the $i$th neuron belongs to module $\rho$. Using this approach numerically, yields nearly identical results as **Equation 52**.

