## [Decision Letter]

Thank you for submitting your article "Velocity coupling of grid modules enables stable embedding of a low dimensional variable in a high dimensional attractor" for consideration by *eLife*. Your article has been reviewed by three peer reviewers, and the evaluation has been overseen by a Reviewing Editor and Laura Colgin as the Senior Editor. The following individual involved in review of your submission has agreed to reveal her identity: Ann M Hermundstad (Reviewer #2).

The reviewers have discussed the reviews with one another and the Reviewing Editor has drafted this decision to help you prepare a revised submission.

Summary:

This paper develops a model in which grid modules are formed by a standard attractor mechanism, and are also coupled together by recurrent interactions to coordinate stochastic drift (caused by noisy velocity inputs and/or spiking noise) between the modules. Independent drift in grid modules would severely impair the fidelity of their spatial representation (as is shown elegantly in this work), so that if grid cells represent space, it is critical to coordinate the drift in modules. The authors show that a simple mechanism – excitatory interaction between modules – can solve this problem, and suggest experimental tests of the theory. In particular, the theory makes interesting predictions arising from the inactivation of a subset of the modules (e.g. self-motion integration in the remaining modules should be affected in a specific way).

Essential revisions:

1) The choice of readout mechanisms might affect the results – hence this key aspect should be better explored.

2) The system's performance and error should be better characterized – as these are the main observables used to draw conclusions about the effectiveness of the proposed model.

Specific points concerning these first two essential revisions (followed by more essential revisions below):

a) It would be good to provide further intuition on the structure of the C_ij that achieved "satisfactory" performance. In particular, even if one constrains the coupling to lie between neighboring modules, the structure of the C_ij was not entirely clear. Is it all-to-all? Does this coupling vary in any interesting way with separation on the neural sheet? Can this coupling be constrained to be somewhat local on the neural sheet and still achieve good results?

b) The manuscript's main argument is that small differences in the phases of individual modules can lead to large errors in the estimate of position that is read out from these modules; by choosing the appropriate coupling between modules, one can reduce deviations in the phases between different modules, and in turn reduce the frequency of large readout errors. It appears that the existence and degree of these discontinuities is due at least in part to the choice of readout. As argued in the framing of the paper and shown in Figures 4D, 5B, these discontinuities are not present in the phases of individual modules, but rather arise in the readout stage. These discontinuities occur within a single timestep (at least as far as one can tell from Figure 5B), suggesting that a readout with a slightly longer timescale could potentially alleviate some of these errors. The form of the readout was derived by the same authors in a previous paper; in that paper, it was shown that the optimal readout should have different timescales for grid modules with different spacings, ranging from 1ms to 600ms. However, here a readout with a single timescale of 10ms was chosen for all modules. Based on this previous work, it's not clear why this choice was made, and what implications this has for the extent of readout errors that are shown in Figure 5B. To make the strongest argument possible, an optimal version of the readout should be used (with appropriately-chosen timescales for different modules) in order to construct the strongest null model in the absence of coupling. As it is now, it's unclear whether the observed readout errors arise from the construction of readout itself, rather than the lack of couplings. Given that this is the primary argument for necessitating couplings between modules, this should be addressed.

c) The paper would be stronger if the analyses provided a better characterization of performance and error. As it's written now, it's difficult to gain an intuition as to why catastrophic errors are occurring when they do, and how much gradual error is accumulated between these catastrophic errors. There are some small fixes that could help (see minor points). Beyond these, there are some larger points that could be made more clear. Why does the first catastrophic error occur when it does (i.e., what is the significance of the timescale marked by the dotted line in Figure 5C-D)? This timescale might have been expected to relate to the distance that must be traveled before the three grid spacings are likely to overlap, given the scale of the noise; is this the case? If so, one would have expected a sharper drop off in the percentage of successes in Figure 5C. If not, is there an intuition as to why the errors occur when they do? Does gradual drift accumulate much more rapidly in the uncoupled model that in the coupled model? This itself is interesting and would be valuable to quantify.

d) Finally, how does performance scale with the number of modules? A general theoretical model was derived for m modules but results are shown for only 2 and 3 modules (and because these two cases were illustrated in one and two dimensions, respectively, it's not possible to compare directly between them). Larger numbers of modules could presumably lead to a couple of different scenarios that might impact the conclusions in different ways: i) more modules could lead to a reduction in readout errors (even in the absence of coupling), because the readout is averaging over inconsistencies among modules; this would suggest that some of observed error could be alleviated with additional modules, or ii) more modules could lead to an increase in readout errors via the additional inter-module recurrence (e.g. in cases where assumptions about linearity or small input velocity break down). It would be useful to report more on these type of analyses and show how they depend on the source/magnitude of noise. Can these findings be interpreted in terms of the actual numbers of modules in mEC (which would be helpful to report somewhere)?

3) Provide, in the main text, more details/intuition about the choice of inter- and intra-module couplings, which is indeed a fundamental aspect of the work. (See also point 1 and 2a above.)

A key point of the theory is that modules should be coupled in a phase-independent manner, and that the coupling should only depend on the rate of change of the phase. There are two things that could be shown, perhaps in Figure 1 and incorporated into Figure 2:

– show the detrimental effect of phase-dependent coupling for position encoding

– recap the properties of self-motion integration networks (e.g. Seung's model used throughout the paper), showing the velocity dependent tuning of the left and right network and how a simple set of synaptic weights between modules can extract this information.

4) Discuss the role of landmarks (e.g. boundaries) in path integration – this important aspect did not receive enough attention throughout the manuscript.

For example, the section on "Intrinsic neural noise" shows how the coupling between modules coordinates their drift. An alternate idea might have been to largely shut down the drift via interaction with border cells (or more generally with boundaries and landmarks). A comment on this alternative would be in order in the Results and in the Discussion since there has been a good bit of work on it lately. These kinds of error correction mechanisms that people have considered so far still work separately on each module, and so a coordination mechanism such as the one in the present paper will still be necessary because even a small amount of independent drift between modules can cause problems with the spatial representation.

5) An additional figure should be added at the end of the paper, illustrating the non-trivial prediction related to the inactivation of one or more grid cell modules. This would help make the work for broadly accessible, and would strengthen the discussion.

6) Discuss the possibility that other (published) mechanisms might be at work for the self-organized emergence of modular structure (which is otherwise assumed in the current manuscript).

In particular, a potential overlap is with recent work on the role of interactions with borders and landmarks in anchoring and reducing drift in the grid system (e.g. Keinath et al., *eLife*; Ocko et al., PNAS; Hardcastle et al., Neuron; Pollock et al., bioRxiv). The present paper is complementary in that it suggests a way of coordinating drift between modules, rather than reducing the overall amount of the drift. Indeed the border interaction mechanisms may not reduce drift enough on their own, and the coordination mechanism described in the present paper may be necessary also – both mechanisms could be in play. Some commentary would be worthwhile in the Discussion as an interesting direction for the future is to ask how border interactions and inter-module interactions could work together.

---

## [Author Response]

Essential revisions:1) The choice of readout mechanisms might affect the results – hence this key aspect should be better explored.2) The system's performance and error should be better characterized – as these are the main observables used to draw conclusions about the effectiveness of the proposed model.Specific points concerning these first two essential revisions (followed by more essential revisions below):a) It would be good to provide further intuition on the structure of the C_ij that achieved "satisfactory" performance. In particular, even if one constrains the coupling to lie between neighboring modules, the structure of the C_ij was not entirely clear. Is it all-to-all? Does this coupling vary in any interesting way with separation on the neural sheet? Can this coupling be constrained to be somewhat local on the neural sheet and still achieve good results?

There are two different ways in which the term ‘all-to-all connectivity’ can be used in relation to our work: one context is when considering how two modules are coupled: every neuron in one module is coupled to every neuron in the other module. A second context has to do with the coupling strengths between modules, quantified byCμρ. We focus first on. Cμρ

The optimization goal that we formulated (Equation 63 and surrounding text) involves only the diagonal elements of the matrix *C*, so it does not determine the non-diagonal elements. Consequently, the optimization yields *m* equations, one for each element on the diagonal. The non-diagonal elements, together with the diagonal elements, are constrained by another set of *m* equations, stating that u→ is a null eigenvector of *C*. Since the number of parameters in *C* is quadratic in m, whereas the number of equations is linear, the solution is highly degenerate for anym>2. This degeneracy means that our optimization procedure cannot determine the structure ofCμρ, but we can seek solutions with additional properties. General solutions involve all-to-all connectivity between modules, but we have chosen to work with a solution in which modules are only coupled to their adjacent modules. This is motivated by the fact that there is evidence for topographic organization of modules on the cortical sheet, and for a limited range of excitatory connections within the MEC (see Discussion). Thus, a solution in which the connectivity is only between adjacent modules may be more compatible (compared to other solutions) with what is currently known about the anatomy. However, the data is not yet conclusive on these questions.

We now write in detail (Appendix 3) the solution for *m* modules, and explicitly discuss the degeneracy of the solution. We then explicitly provide the matrix *C* obtained under the additional restriction that only adjacent modules are coupled (Equation 75).

When considering the coupling between neurons in any two modules, the connectivity is all-to-all in the sense that every neuron in one module is synaptically coupled to every neuron in the other module (assuming that the corresponding Cμρ is non-vanishing). This form of connectivity is central to our proposal, as it entails complete symmetry to shifts in positions of the activity bumps in the two modules. The symmetry ensures that the set of steady states will involve any combination of bump positions, thereby preserving the large capacity of the code. This is now clarified in subsection “Coupling modules by synaptic connectivity”.

b) The manuscript's main argument is that small differences in the phases of individual modules can lead to large errors in the estimate of position that is read out from these modules; by choosing the appropriate coupling between modules, one can reduce deviations in the phases between different modules, and in turn reduce the frequency of large readout errors. It appears that the existence and degree of these discontinuities is due at least in part to the choice of readout. As argued in the framing of the paper and shown in Figures 4D, 5B, these discontinuities are not present in the phases of individual modules, but rather arise in the readout stage. These discontinuities occur within a single timestep (at least as far as one can tell from Figure 5B), suggesting that a readout with a slightly longer timescale could potentially alleviate some of these errors. The form of the readout was derived by the same authors in a previous paper; in that paper, it was shown that the optimal readout should have different timescales for grid modules with different spacings, ranging from 1ms to 600ms. However, here a readout with a single timescale of 10ms was chosen for all modules. Based on this previous work, it's not clear why this choice was made, and what implications this has for the extent of readout errors that are shown in Figure 5B. To make the strongest argument possible, an optimal version of the readout should be used (with appropriately-chosen timescales for different modules) in order to construct the strongest null model in the absence of coupling. As it is now, it's unclear whether the observed readout errors arise from the construction of readout itself, rather than the lack of couplings. Given that this is the primary argument for necessitating couplings between modules, this should be addressed.

The decoder that we use in the present manuscript was developed in Mosheiff et al., 2017, and its MSE in each module was quantified as a function of the time scale of readout. The reason why we used a time scale of 10 ms is that with the parameters that we use in the present work (see below), the optimal time scale for readout is very short (smaller than 1ms in all three modules), whereas readout on a time scale of 10 ms seems more easy to implement biologically.

The comment above raises an interesting concern, that the time scale of readout might influence the rate at which catastrophic readout errors occur, and thereby significantly influence the MSE of readout. To address this question, we repeated the analysis with different readout time scales, both shorter and longer than 10ms (1ms, 10ms, and 100ms). The results are shown in a new supporting figure (Figure 5—figure supplement 1), and discussed briefly in paragraph three of subsection “Consequences for spatial representation and readout”. We find that the time scale of readout has very little influence on occurrence of catastrophic readout errors (Figure 5—figure supplement 1A), or on the qualitative behavior of the MSE curves (Figure 5—figure supplement 1B). Therefore, the choice of the readout time scale is not essential in the context of the present work.

The main goal of Mosheiff et al., 2017 was to propose a normative explanation for a trend seen in the experimental data (Stensola et al., 2012), according to which modules with larger spacing appear to include a smaller number of neurons. However, it is not yet clear whether this trend is representative of the true distribution of cells in the MEC, or whether it was a result of an experimental bias. We think that in the present work, that addresses different questions, it makes more sense to make the simple choice of identical number of neurons in each module, which is why the optimal time scales for readout are small in all modules.

c) The paper would be stronger if the analyses provided a better characterization of performance and error. As it's written now, it's difficult to gain an intuition as to why catastrophic errors are occurring when they do, and how much gradual error is accumulated between these catastrophic errors. There are some small fixes that could help (see minor points). Beyond these, there are some larger points that could be made more clear. Why does the first catastrophic error occur when it does (i.e., what is the significance of the timescale marked by the dotted line in Figure 5C-D)? This timescale might have been expected to relate to the distance that must be traveled before the three grid spacings are likely to overlap, given the scale of the noise; is this the case? If so, one would have expected a sharper drop off in the percentage of successes in Figure 5C. If not, is there an intuition as to why the errors occur when they do? Does gradual drift accumulate much more rapidly in the uncoupled model that in the coupled model? This itself is interesting and would be valuable to quantify.

We added a panel in Figure 4F, quantifying how relative errors between modules accumulate over time, averaged over multiple trials. The parameters are the same as those used in Figure 5. This panel is discussed in paragraph five of subsection “Generalization to two dimensions and several modules”.

The role of the vertical dotted line in Figures 5C-D is to point out the strong relationship between large MSE and the occurrence of catastrophic readout errors. However, the time at which the first catastrophic error occurs depends not only on dynamic properties of the system, but also on the number of simulations performed: in our case, it roughly corresponds to the time at which the probability for occurrence of a catastrophic error is of order 1%, since we ran 100 simulations. To the left of the dashed line the probability to have a catastrophic error does not vanish, it’s simply very small. This is now clarified in the text (paragraph three subsection “Consequences for spatial representation and readout”).

Catastrophic errors due to relative phase shifts have been extensively discussed in the literature by us and others (Fiete et al., 2008, Wellinder et al., 2008, Sreenivasan and Fiete, 2011, Burak, 2014, Vágó and Ujfalussy, 2018). However these works focused on the capacity of the code and its resolution. A quantitative analysis of the probability for occurrence of catastrophic readout errors, and how this probability scales with various parameters, has not been achieved. We feel that this question is interesting but lies outside the scope of our manuscript for two main reasons:

1. There is no question that these errors are highly detrimental (see citations mentioned above).

2. Our goal is to propose a mechanism that mitigates these errors, and we demonstrate clearly that our proposed mechanism achieves this goal.

As a basic and rough argumentation, it seems reasonable that in order for a catastrophic readout error to occur, relative phase shifts between modules should become significant compared to unity. Based on the new panel (Figure 4F), it is possible to see that when catastrophic errors started to appear in Figure 5C, the standard deviation of phase shifts was of order 0.15 (MSD ~ 0.025), and there was already a small but appreciable probability for relative phase shifts to become of order unity.

However, based on previous works (Fiete et al., 2008, Vágó and Ujfalussy, 2018), it should be clear that the probability for a catastrophic readout must depend in a complicated manner on other factors such as the precise grid spacings, the number of modules and the size of the arena. The new supporting figure (Figure 5—figure supplement 2) demonstrates some of this complexity.

We added a paragraph in the Discussion section (paragraph three) addressing some of the points raised above.

Regarding the last question: we added a quantification of the mean square error relative to the true trajectory, Figure 4G. In the coupled system, Noise in the three modules is effectively projected on the direction corresponding to joint motion, which is equivalent to averaging the noise injected to each module individually. The mean square error accumulated in each module individually is thus expected to be smaller by a factor of *m* in the case of coupled modules, compared to the uncoupled case, in agreement with our numerical results. We explain this in paragraph five of subsection “Generalization to two dimensions and several modules”.

d) Finally, how does performance scale with the number of modules? A general theoretical model was derived for m modules but results are shown for only 2 and 3 modules (and because these two cases were illustrated in one and two dimensions, respectively, it's not possible to compare directly between them). Larger numbers of modules could presumably lead to a couple of different scenarios that might impact the conclusions in different ways: i) more modules could lead to a reduction in readout errors (even in the absence of coupling), because the readout is averaging over inconsistencies among modules; this would suggest that some of observed error could be alleviated with additional modules, or ii) more modules could lead to an increase in readout errors via the additional inter-module recurrence (e.g. in cases where assumptions about linearity or small input velocity break down). It would be useful to report more on these type of analyses and show how they depend on the source/magnitude of noise. Can these findings be interpreted in terms of the actual numbers of modules in mEC (which would be helpful to report somewhere)?

We agree that this is an interesting question. The actual number of grid cell modules in the brains of rats and mice is unknown, and so far there is direct evidence for the existence of four modules.

We added a supporting figure (Figure 5—figure supplement 2) with several panels that show how the probability for catastrophic errors, and the MSE, evolve in time when the number of modules is increased. We find that increasing the number of modules reduces the probability for catastrophic errors, in accordance with the intuition described above under scenario i). In interpreting these figures, however, it is important to keep in mind that the size of the environment is important. Increasing the number of modules leads to an increase in the range of positions that can be represented unambiguously. Therefore, when comparing the rate of catastrophic readout errors across systems that differ in the number of modules, one has to decide whether to keep the size of environment fixed, or increase it to reflect the increased capacity. Increasing the size of the environment goes hand-in-hand with an increase in the probability for catastrophic errors, since the larger environment includes a larger set of positions whose representation by the grid phases might be confused with the true position. This is demonstrated in panels E-J in Figure 5—figure supplement 2.

In fact, spatial coding and memory maintenance in the entorhinal-hippocampal system may require resilience to an even broader range of errors than those associated with confusion between two positions in any given environment. An error in module phases, accumulated during foraging in the absence of sensory cues, might lead to a combination of phases that matches a position in a different spatial map (representing a different environment), more than any position in the present map.

The main goal of the examples shown in Figure 5 and the new Figure 5—figure supplement 2 is not to systematically analyze the effect of catastrophic errors. The goal is to demonstrate that these effects are highly detrimental over a broad range of parameters, and that the coupling mechanism proposed in our work dramatically reduces these detrimental effects.

We briefly discuss Figure 5—figure supplement 2 in the text (paragraph four of subsection “Consequences for spatial representation and readout”), and have added a paragraph in the Discussion section addressing some of the issues raised above (paragraph four).

3) Provide, in the main text, more details/intuition about the choice of inter- and intra-module couplings, which is indeed a fundamental aspect of the work. (See also point 1 and 2a above.)A key point of the theory is that modules should be coupled in a phase-independent manner, and that the coupling should only depend on the rate of change of the phase. There are two things that could be shown, perhaps in Figure 1 and incorporated into Figure 2:– show the detrimental effect of phase-dependent coupling for position encoding– recap the properties of self-motion integration networks (e.g. Seung's model used throughout the paper), showing the velocity dependent tuning of the left and right network and how a simple set of synaptic weights between modules can extract this information.

We added to Figure 1 a schematic illustration of two phase-coupled gears (panel C), and included a short explanation inside the figure caption on why this scheme is unfavorable from the point of view of capacity. We refer to this point also in paragraph six of the Introduction.

We also added a panel to Figure 2 (panel B), illustrating the difference in activity in the left and right bumps as the source of motion along the attractor, and expanded the text in the caption to explain how the double ring model works. Note that the section after Equation 3 in the main text also explain the mechanism of velocity integration in the double ring model.

The readout mechanism that we propose to each module’s velocity is derived and discussed extensively in Appendix 1. We now also explain in the main text (paragraph two of subsection “Coupling modules by synaptic connectivity”) and in the Materials and methods section (after Equation 13) in more detail the structure of synaptic connectivity that injects this readout velocity (as read out from each module) as input to the other modules.

4) Discuss the role of landmarks (e.g. boundaries) in path integration – this important aspect did not receive enough attention throughout the manuscript.For example, the section on "Intrinsic neural noise" shows how the coupling between modules coordinates their drift. An alternate idea might have been to largely shut down the drift via interaction with border cells (or more generally with boundaries and landmarks). A comment on this alternative would be in order in the Results and in the Discussion since there has been a good bit of work on it lately. These kinds of error correction mechanisms that people have considered so far still work separately on each module, and so a coordination mechanism such as the one in the present paper will still be necessary because even a small amount of independent drift between modules can cause problems with the spatial representation.

This is an important point which was addressed in our initial submission only indirectly, by highlighting the significance of coupling under conditions in which sensory cues are absent. We added a paragraph in the Discussion section (paragraph six), in which we comment on the role of boundaries in this context, and cite recent related work. We also briefly mention the possible role of encounters with the walls as a means for updating the spatial representation in paragraph four of subsection “Generalization to two dimensions and several modules”.

5) An additional figure should be added at the end of the paper, illustrating the non-trivial prediction related to the inactivation of one or more grid cell modules. This would help make the work for broadly accessible, and would strengthen the discussion.

Thanks for this suggestion. We added a figure (Figure 4—figure supplement 1) showing the rate maps of individual neurons from three modules after disconnecting module 1 from the other modules. We refer to this figure in subsection “Experimental predictions”.

6) Discuss the possibility that other (published) mechanisms might be at work for the self-organized emergence of modular structure (which is otherwise assumed in the current manuscript).

See added text and citation in paragraph seven of the Discussion.

In particular, a potential overlap is with recent work on the role of interactions with borders and landmarks in anchoring and reducing drift in the grid system (e.g. Keinath et al., eLife; Ocko et al., PNAS; Hardcastle et al., Neuron; Pollock et al., bioRxiv). The present paper is complementary in that it suggests a way of coordinating drift between modules, rather than reducing the overall amount of the drift. Indeed the border interaction mechanisms may not reduce drift enough on their own, and the coordination mechanism described in the present paper may be necessary also – both mechanisms could be in play. Some commentary would be worthwhile in the Discussion as an interesting direction for the future is to ask how border interactions and inter-module interactions could work together.

See the paragraph six in the Discussion.